# Review on Additives in Hydrogels for 3D Bioprinting of Regenerative Medicine: From Mechanism to Methodology

**DOI:** 10.3390/pharmaceutics15061700

**Published:** 2023-06-09

**Authors:** Wenzhuo Fang, Ming Yang, Meng Liu, Yangwang Jin, Yuhui Wang, Ranxing Yang, Ying Wang, Kaile Zhang, Qiang Fu

**Affiliations:** Department of Urology, Affiliated Sixth People’s Hospital, Shanghai Jiaotong University, No. 600 Yi-Shan Road, Shanghai 200233, China; fwz17794608029@163.com (W.F.); yangminguro@163.com (M.Y.);

**Keywords:** 3D bioprinting, hydrogel, bioink, tissue engineering, bionic scaffold

## Abstract

The regeneration of biological tissues in medicine is challenging, and 3D bioprinting offers an innovative way to create functional multicellular tissues. One common way in bioprinting is bioink, which is one type of the cell-loaded hydrogel. For clinical application, however, the bioprinting still suffers from satisfactory performance, e.g., in vascularization, effective antibacterial, immunomodulation, and regulation of collagen deposition. Many studies incorporated different bioactive materials into the 3D-printed scaffolds to optimize the bioprinting. Here, we reviewed a variety of additives added to the 3D bioprinting hydrogel. The underlying mechanisms and methodology for biological regeneration are important and will provide a useful basis for future research.

## 1. Introduction

Owing to the rapid development of current medical technology, a high success rate in organ transplantation and tissue repair was recently achieved. However, the limited number of organs available from donors constrain organ transplantation and tissue repair [1]. In recent years, researchers focused on tissue engineering and regenerative medicine to overcome donor shortage and immune rejection. Tissue engineering focuses on creating autologous tissues or organs for transplantation by using cells, biomaterials, and some additives such as cytokines. Additionally, regenerative medicine reproduces the healthy functions of cells, tissues, or organs through substitution or regeneration [2,3].

An emerging technology is 3D printing; it made significant advances in many fields, including aerospace, consumer products, the food industry, and manufacturing, and it combines cells and biological materials [4] to construct complex 3D structures, which can be used for scaffold-based or scaffold-free tissue and organ structures, micro-organisms, and single-chip organ model systems. For the medical field, 3D printing is also revolutionary in treating diseases [3,5,6,7]. The core of 3D printing lies in the printing conditions, printing technology, and suitable bioinks. Bioinks are generally a formula containing biomaterials and bioactive components that can be processed by automated manufacturing technology [8,9]. The types of bioinks include hydrogels, cell aggregates, acellular matrix, etc. [10,11]. The ideal hydrogel materials should have good physical, chemical, and biological characteristics, such as printability, biocompatibility, degradability, mechanical properties, stability, non-toxic, and non-immunogenic, as well as promotion of cell adhesion proliferation and differentiation [11,12,13]. However, possession of these properties does not necessarily permit the functional scaffolds to achieve optimal tissue repair, as the microenvironment and growth conditions at various implantation sites are different. Problems such as insufficient vascularization, causing ischemia and hypoxia, microbial contamination, inflammatory reactions, collagen metabolism disorders, etc., may occur after scaffold implantation into the target tissue. Scaffold implantation, tissue regeneration, wound healing, and scar formation will then fail. Therefore, it is necessary to design the specific scaffold to better enhance the regeneration-promoting ability [14,15]. In this review, we summarize different additives that were added to 3D bioprinting bioinks. As mentioned above, we divided the various additives into these four categories precisely because sufficient vascularization, lower inflammatory response and infection, and appropriate collagen deposition are key to better adaptation of the repair site to the implant and are among the indispensable factors to improve implant success. As shown in Table 1, we first describe, in general terms, the functions of the different additives and the mechanisms by which they work, followed by examples of the specific ways in which different researchers applied them to hydrogel scaffolds and the good results that were eventually achieved. We believe that the major difference and advantage of this review is that it has the potential to be a timely report for future researchers in selecting the right additives for bioinks. By understanding the functions of different additives and the mechanisms by which they function, as well as by referring to the methodologies of others, it will help researchers to find the additives they are looking for more efficiently and conveniently.

## 2. Vascularization

Good vascularization and vascular recanalization is crucial for the repair sites because blood vessels are vital to tissue repair [44,45]. Blood vessels can provide sufficient oxygen and nutrients and carry away metabolic wastes [46,47,48,49]. Therefore, the purpose of 3D printed scaffolds is to encourage vascular regeneration, which leads to in-tissue repair and regeneration. The details to make scaffolds are given below.

### 2.1. Growth Factors

In recent years, growth factors are being widely noticed. Specific cell receptors respond differently after being stimulated by growth factors [50,51]. Adding growth factors to the 3D bioprinting process can stimulate cell proliferation and differentiation, thus increasing cell activity and increasing the success rate of 3D bioprinting [52]. The activity of endothelial cells is additionally crucial for successful vascularization. The process of endothelium-mediated angiogenesis is complex and is impacted by multiple growth factors. Understanding the function of various growth factors and their methods for promoting angiogenesis is crucial for tissue repair. The growth factors that used in 3D bioprinting for vascularization such as vascular endothelial growth factor (VEGF) are listed below.

#### 2.1.1. VEGF

VEGF is a vascular endothelial cell-specific heparin-binding growth factor that can induce vascular neogenesis in vivo and is a highly conserved homodimeric glycoprotein with a high specificity for cell mitosis. In vivo, VEGF induces endothelial cell proliferation, promotes cell migration, and inhibits apoptosis. In contrast, VEGF-induced angiogenesis with increased vascular permeability plays a critical regulatory role in the generation of blood vessels [53,54]. Both the short half-life of VEGF and the lack of a suitable long-term sustained release device make it difficult to ensure effective angiogenesis as well as raise waste and safety issues [3]. Therefore, it is necessary to select a suitable controlled delivery system.

Encapsulation of growth factors into microspheres before incorporating into 3D printed scaffolds was shown to be one of the most effective ways to achieve controlled release [55,56]. Wang et al. [16] made bone morphogenetic protein-2 (BMP-2) and VEGF adsorbed onto silk fiber (SF) microspheres (diameter of 1.5 ± 0.3 μm) that were prepared using a co-flow capillary device. These microspheres were, subsequently, doped into the SF/nano-hydroxyapatite (nHAp) scaffolds to provide regulated release. As shown in Figure 1, BMP-2 and VEGF are incorporated into SF microspheres by chemical covalent bonding via EDC/NHS (1-ethyl-3-(3-(dimethylamino) propyl) carbodiimide (EDC) and N-hydroxysuccinimide (NHS)) chemistry and physisorption, respectively, which leads to the controlled and sustained release from the SF/nHAp scaffold. The initial rapid release of VEGF mimics its expression during the early bone healing phase and promotes angiogenesis. The comparatively gradual and continuous release of BMP-2 facilitated osteogenic differentiation both in vitro and in vivo.

After 12 weeks of inserting the printed scaffold into the rat cranial defect location, a CD31 staining experiment showed that blood vessels were identified as brown, round, or oval structures. Compared with control or BMP-2 scaffolds, scaffolds that were loaded with VEGF showed a much higher blood vessel density, and the VEGF group exhibited the strongest signal among all groups. These findings suggested that the scaffolds that contained VEGF help vascularization. Furthermore, compared with BMP-2 or VEGF alone, the combination of BMP-2 and VEGF produced a synergistic effect on bone formation in rat cranial defects with good scaffold angiogenic properties and better osteogenesis, since vascularization is crucial for bone regeneration [57,58].

#### 2.1.2. FGF-2

Fibroblast growth factor-2 (FGF-2) is a multifunctional peptide that promotes the growth, migration, and differentiation of many cell types [59]. FGF-2 was also shown to stimulate VEGF expression, granulation tissue formation, and vascular maturation [60,61]. Xiong et al. [17] used 3D printing to fabricate a gelatin-sulfonated silk composite scaffold (3DG-SF-SO_3_-FGF). The basic FGF-2 was incorporated in this scaffold by binding with a sulfonic acid group (SO_3_). SF has good biocompatibility, mechanical properties, stability, and non-inflammatory response and can be used as a repair for skin, bone, blood vessels, and other tissues [62,63,64]. Modified SF with SO_3_ was used in this scaffold to increase its hydrophilicity and can promote the binding of FGF-2. The in vitro release profile of FGF-2 from 3DG-SF-SO_3_-FGF scaffold showed that a 32% burst release was detected during the first 24 h, followed by a sustained slow release profile. A total of 30% of the FGF-2 remained in the stent after 12 days. Then, 4 weeks after implantation, skin wounds of rats showed significantly enhanced regeneration of dermatomal tissue due to the addition of FGF-2. Further investigation showed that a more extensive distribution of vascular recanalization was observed in the scaffold group with FGF-2 binding. Additionally, immunohistochemistry showed a significant increase in the expression of vascular-associated proteins such as α-smooth muscle actin (α-SMA) and CD31. These results demonstrate the importance of FGF-2 in recruiting endogenous cells for epithelialization and angiogenesis at defective sites.

#### 2.1.3. PDGF

Some other growth factors were found to promote angiogenesis, e.g., platelet-derived growth factor (PDGF), which can recruit smooth muscle cell (SMC)/pericytes to immature vessels to stabilize and remodel them [65]. The deficiency of this factor will lead to the maintenance of a proliferative endothelial cells (ECs) phenotype, which is detrimental to maturation and even causes vascular degeneration. Currently, there are few experiments in applying them alone to 3D printing technology for tissue regeneration and angiogenesis.

### 2.2. Heparin and Its Derivatives

Heparin is a negatively charged polysaccharide macromolecule that has good affinity for angiogenic factors. Nih et al. proved that it can accelerate neovascularization by binding angiogenic factors, such as VEGF, and improving their stability [66,67,68,69]. An et al. [18] coated poly-L-lysine (PLL) and heparin on the surface of 3D-printed hydrogel scaffolds, which were made from cross-linking of gelatin methacrylate (GelMA) and hyaluronic acid methacrylate (HAMA) via electrostatic interactions between PLL (positively charged) and heparin (negatively charged). Then, they were inoculated with hypoxia-inducible factor-1α (HIF-1α) mutated muscle-derived stem cells (MDSCs). At last, the scaffolds were implanted into the injured cavernous body of rabbits finally. After 4 months, the erectile and ejaculatory functions of rabbit’s penile corpus cavernosum were greatly improved. Additionally, compared with the non-heparin-coated group, the relative protein expression of angiogenic markers (VEGF, PDGF, and stroma cell-derived factor-1 (SDF-1)) was significantly increased in the heparin-coated group. Specifically, angiogenic factors produced by HIF-1αmutant MDSCs were accumulated on the scaffold surface by adsorption of heparin, which, subsequently, stimulated angiogenesis in vivo. The increase in neovascularization further promoted the recovery of erectile and ejaculatory function in the injured penile corpus cavernosum.

Heparan sulfate (HS), one of the heparin derivatives, is a linear polysaccharide of the glycosaminoglycan family, which can be synthesized by almost all animal cells [70]. It has good mechanical properties and thermal stability, and has a large number of binding sites for bioactive molecules. In addition, HS can not only protect growth factors from degradation by proteases [71,72], but it also promote the binding of growth factors to their receptors, thus promoting growth factor activity [73]. Jiang et al. [19] fabricated a scaffold with collagen and HS and implanted it at the site of traumatic brain injury in hounds. In situ hybridization (ISH) and immunofluorescence staining showed that HS-containing scaffold promoted vascular and nerve regeneration compared with the control group, and that micro-vessel formation may underlie tissue regeneration and nerve repair.

### 2.3. Ionic Composition

Many studies incorporated various ionic components with osteogenic, angiogenic, and other functions directly into scaffold materials to allow the controlled delivery, or to functionalize the scaffold surface, or to make scaffolds have specific biological functions [74,75,76]. In addition, most of these ions have some biological interactions with cells (e.g., promoting the expression of growth factors by endothelial cells, promoting cell migration differentiation and proliferation, etc.), and perform pro-vascularization functions via these interactions. Table 2 shows the role of the different ions in the hydrogel and briefly describes the mechanisms by which they function and how they are applied.

#### 2.3.1. Silicon Ions

Silicon (Si) is one of the essential elements for the development of healthy bones and blood vessels [86,87,88]. It was demonstrated that the addition of Si ions may promote osteogenic differentiation of human bone marrow stromal cells (hBMSCs) [89], and play a significant role in enhancing angiogenesis of ECs [90,91]. Many researchers found that the expression of many factors related to angiogenesis (e.g., VEGF, kinase insert domain-containing receptor (KDR), and HIF-1α et al.) in vascular endothelial cells was significantly boosted under the stimulation of Si ions. In addition, in vitro migration, differentiation, and tubular formation of vascular endothelial cells were stimulated in the presence of Si ions. To better utilize the pro-vascularization and osteogenic functions of Si ions, Li et al. [77] used a novel calcium phosphate cement (CPC) as the basis for a scaffold that combined mesoporous silica (MS) with recombinant human bone morphogenetic protein-2 (rhBMP-2) to promote osteogenesis and angiogenesis. This bioactive mesoporous scaffold has well-interconnected macropores. Due to the inherent self-setting properties of calcium phosphate cements, the scaffolds can be printed at room temperature without sintering afterwards and have enhanced mechanical strength. This study showed that more newly formed vessels could be observed in the MS/CPC group compared to the CPC-only group. As seen in Figure 2, the micro-computerized tomography (μCT) results of the rabbit radius—4 weeks after stent implantation—showed that abundant blood vessels formed around the CPC-only stent, and no blood vessels were observed inside the stent. The MS/CPC and MS/CPC/rhBMP-2 stents not only had abundant blood vessels around the stent, but the blood vessels also grew along the interconnected macropores inside the stent. This suggests significant promotion of vascularization by Si ions. However, this study did not delve into the mechanism of Si ions on promoting neovascularization, and so, further studies are needed in this area.

#### 2.3.2. Magnesium Ions

Magnesium (Mg) is one of the essential elements of the human body and is associated with many physiological activities [92,93]. Compared to other metallic or polymeric materials, Mg can promote vascular growth and improve local blood perfusion [94,95], they also play a direct and important role in maintaining vascular function [96], especially for vascular endothelial cells. It is well known that vascular endothelial cells play an irreplaceable role in angiogenesis and that capillary branching, germination, and proliferation are dependent on them. It was shown that magnesium ions stimulate the proliferation of HUVEC and increase their response to some motogenic factors. In addition, Mg ion can also upregulate integrin function, which is one of the reasons why it can promote endothelial cell migration [94,95]. In addition, Mg ions were also demonstrated to have the ability to promote new bone formation and bone growth [97,98].

Gu et al. [78] fabricated Mg-doped β-TCP (Mg-TCP) scaffolds by 3D printing and sintering, in which MgO was mixed in four groups at 0, 1, 2, and 4 wt% ratios. β-TCP, as one of the commonly used biomaterials for scaffold preparation, has excellent biocompatibility and biodegradability, and its composition is also similar to that of natural bone minerals [99]. Endothelium-derived nitric oxide (NO) was used as an index to evaluate the angiogenic capacity. In vitro experiments showed that the 1Mg-TCP group produced the most NO compared to the other groups of human umbilical vein endothelial cells (HUVECs) cultured in extracts. This indicated that Mg^2+^ had the strongest angiogenic capacity at this dose, but the specific correlation between the high and low angiogenic capacity and Mg^2+^ dose was not elucidated.

Due to the inherent properties of some scaffolds, Mg^2+^ can’t be immobilized on its inner and outer surfaces uniformly. Therefore, it should rely on other substances to assist in immobilization, such as polydopamine (PDA). Inspired by mussel adhesion, PDA is able to hold onto almost anything and was extensively studied in many surface-modification experiments [100,101]. Ma et al. [79] used the surface adhesion ability of PDA to make Ta-PDA-Mg scaffolds by doping Mg^2+^ on the surface of 3D printed tantalum scaffolds. Then, Mg(NO_3_)_2_ was added to the surface of the scaffolds at doses of 0, 10, 20, and 50 mg, and the four groups of scaffolds were named Ta-PDA, Ta-PDA-Mg1, Ta-PDA-Mg2, and Ta-PDA-Mg3, respectively. The results showed that the constructed Ta-PDA-Mg scaffold promoted osteogenic and angiogenic responses in vitro and in vivo compared with the control group. In particular, Ta-PDA-Mg2 is the most effective (not Ta-PDA-Mg3), and immunofluorescence showed more mature angiogenesis in this group.

There is no doubt that Mg^2+^ promotes blood vessel regeneration, but high concentrations of Mg^2+^ may have the opposite effect [78,79]. The decreased angiogenic capacity may be related to the decrease in cell viability caused by high concentrations of Mg^2+^, but most experiments did not involve further studies. In addition, there are few reports on the relationship between the expression of angiogenesis-related genes in endothelial cells and Mg^2+^ concentration.

#### 2.3.3. Copper Ions

Copper (Cu) is one of the indispensable trace elements for human beings, and the appropriate amount of Cu^2+^ is very important for maintaining health [82]. In addition, Cu^2+^ can stimulate endothelial cell proliferation and differentiation. They promote angiogenesis by mimicking hypoxia, stabilizing the expression of HIF-1α, and promoting the expression of VEGF [102,103,104]. Tao et al. [82] fabricated a novel metal-organic framework, a β-tricalcium phosphate (Cu-TCPP-TCP) scaffold containing a Cu^2+^ coordinated tetrakis (4-carboxyphenyl) porphyrin (Cu-TCPP) nanosheet interfacial structure, by using 3D printing technology. Cu-TCPP belongs to the family of metal organic frameworks (MOFs). MOF-based materials can be used for tumor therapy by converting infrared light into thermal energy in the biomedical field [105,106]. It turned out that Cu-TCPP-TCP scaffold supported the attachment of hBMSCs and HUVECs. It also significantly stimulated the expression of genes related to osteogenic differentiation in hBMSCs and angiogenic differentiation in HUVECs. Studies in vivo revealed that after implantation of the Cu-TCPP-TCP scaffold into femoral defects in rabbits, the scaffold effectively promoted regeneration of the femur. So, Cu^2+^-stimulated angiogenesis is essential for the formation of new bone tissue and plays an important role in determining the success or failure of implanted bone tissue grafts [107]. In conclusion, osteogenic bioactivity of the TCP scaffold, photothermal properties of Cu-TCPP nanosheets, and angiogenic activity of Cu^2+^ all enable the multifunctional properties of the Cu-TCPP-TCP scaffold.

### 2.4. Vascular-Promoting Cell

#### 2.4.1. Human Umbilical Vein Endothelial Cells

HUVECs are one of the stem cells derived from umbilical cord tissue who play an important role in angiogenesis that can prevent the need for pre-formed channels or growth factor-induced angiogenesis [108]. It is one of the most popular endothelial cell lines and is widely used to study, thanks to low cost, ease of isolation, and strong angiogenic potential [109]. Cidonio et al. [110] fabricated a 3D scaffold printed with a bioink based on Laponite (LAP) nanoclay and, subsequently, seeded with human umbilical vein endothelial cells and loaded with VEGF. LAP nanoclay, the nanocomposites, can be used as a bioink to make hydrogels to maintain cell activity and create functional 3D implants [111]. Seven days after implantation of the scaffold into the chick chorioal lantoic membrane (CAM), the 3D-printed scaffold, which was equipped with VEGF and HUVECs, was extensively bound to the CAM. In addition, neovascularization penetrated the pores of the printed structures with numerous capillaries around the scaffolds. Moreover, the combination of VEGF and HUVECs had a significantly enhanced effect compared to the individual loading VEGF or HUVECs. This suggests that the interaction of VEGF and HUVECs can be mutually optimized to reduce vascular leakage caused by excessive stimulation of vascular permeability by VEGF [112].

#### 2.4.2. Mesenchymal Stem Cells

Mesenchymal stem cells (MSCs) are a group of adult stem cells of mesodermal origin that can be derived from the placenta, liver, bone marrow, umbilical cord blood, adipose tissue, and heart [113]. MSCs in adult tissues often work as progenitor cells because they are highly proliferative and can differentiate into specific lineages, such as bone cells, endothelial cells, etc. They are considered to be one of the most appropriate cell sources for tissue engineering [114,115].

Although it was demonstrated that cartilage tissue, printed with bioink that contains MSCs, is well-vascularized in vivo because of its capability to secrete a variety of vascular growth factors such as VEGF, FGF-2, and PDGF et al., angiogenesis is often limited in peripheral areas, with no persistent vascularity in the core area of the cartilage [116]. Since angiogenesis plays a crucial role in bone regeneration [21,107,117], it is important to design a 3D-printed scaffold with a controlled internal structure that can guide the repair and vascularization of the core region within the cartilage. Daly et al. [118] used Pluronic ink as a sacrificial bioink to form a network of interconnected microchannels within a GelMA hydrogel, which was filled with MSCs to ensure vascularization of the core region. Four weeks after implantation of the scaffold with microchannel into the bone defect, μCT angiography showed a significant increase in the level of vascularization in the core region of the defect. This design provides the possibility to solve the difficulty of blood vessel formation within cartilage and other tissues.

In addition, a system of co-culture of osteoblasts and angiogenic cells was proposed, which can achieve complementary osteogenesis and angiogenesis [119]. For example, Rong et al. [120] chose induction media to co-culture bi-directionally induced human umbilical cord mesenchymal stem cells (hUCMSCs). HUCMSCs are one of the MSCs that originate from the umbilical cord of newborns. It can secrete cytokines which are related to processes such as inflammation, immune regulation, angiogenesis, and wound healing [121]. The os-hUCMSCs (osteogenically induced hUCMSCs) and en-hUCMSCs (angiogenically induced hUCMSCs) were pre-induced for 3 days before co-culture was performed. In co-culture, pre-induced os-hUCMSCs and en-hUCMSCs were mixed in different ratios and, subsequently, doped into 3D-printed TCP scaffolds. Loading the scaffolds into the defective part of rat skull for 4 weeks showed that the scaffolds had the best osteogenic ability in vitro and in vivo, where the ratio of os-hUCMSCs and en-hUCMSCs was 3:1. The higher ratio the mixed en-hUCMSCs was, the more vascular tubules were formed on the matrix gel. However, this experiment was performed to guide the clinical application of hUCMSCs by directly using human-derived cells in non-immunodeficient SD rats and without postoperative immunosuppression. So, this process will inevitably cause immune rejection. In addition, the genome and proteome of bidirectionally differentiated hUCMSCs need to be further investigated.

In addition to pro-vascularization properties, MSCs in hydrogels also possess immunomodulatory as well as regulatory collagen deposition capabilities. Table 3 lists the mechanisms by which HUVECs and MSCs function in hydrogels and briefly depicts the ways in which researchers have applied them to hydrogels.

### 2.5. Other Materials

Some other additives were demonstrated to exert a certain degree of pro-vascularization.

#### 2.5.1. Desferrioxamine

Desferrioxamine (DFO) is an iron chelator that can bind and assist in the removal of iron ions, and is often used as a hypoxia-mimetic compound under oxygenated conditions. HIF-1α, which is produced during hypoxia, acts as a cellular transcription factor, and was demonstrated to be essential for the regulation of genes related to angiogenesis and osteogenesis [123,124,125]. When DFO was activated, it would lead to the activation of a cascade of pro-angiogenic genes, such as VEGF. Thus, DFO can promote both angiogenesis and bone regeneration when used in fracture models [126,127,128]. Yan et al. [21] designed a bionic degradable polycaprolactone (PCL) scaffold using 3D printing technology. The printed PCL scaffold has macropore and multidirectional channel structure. PCL is one of the most commonly polymers that is used in 3D-printed scaffolds with good biocompatibility and biodegradability. It can control the release of DFO by surface degradation and layer-by-layer assembly techniques [129,130]. Results in vitro showed that the scaffold enhanced the angiogenic activity of HUVECs and the osteogenic activity of BMSCs. Studies in vivo indicated that after implantation of the scaffold at the site of femoral defects in rats, it enhanced rapid vascular invasion and accelerated osseointegration. Overall, DFO gave the scaffold consistent ability to promote angiogenesis and osteogenesis in vitro and in vivo. However, the sample size of this study was too small and a larger sample size is needed to validate the results.

#### 2.5.2. Catalase

It was demonstrated that H_2_O_2_ is produced by cells in a hypoxic state and will contribute to an environment of oxidative stress and ischemia [131,132]. Catalase can break down hydrogen peroxide and produce oxygen, which not only reduces the damage caused by H_2_O_2_, but the oxygen that it produces also helps to induce angiogenesis [133]. Rija et al. [22] intercalated catalase in hydrogels to form functional decellularized adipose tissue-alginate (DAT-A) hydrogels using 3D printing technology. Figure 3 shows the abbreviated manufacturing process.

DAT was widely used in research and clinical applications due to its excellent biocompatibility and degradability. Its main components are collagen and sulfated glycosaminoglycans (GAG) containing hyaluronic acid (HA) [134,135]. In addition, DAT was shown to enhance the extracellular matrix (ECM) and cells, cell–cell interactions, and stem cell differentiation [136]. After implanting the scaffold subcutaneously in rats for 4 weeks, tissue growth increased at the implantation site (≥45%), inflammation reduced (≥40%), and induction in angiogenesis increased (≥40%), as compared to the control group. This demonstrated that the scaffold could successfully promote angiogenesis and tissue regeneration.

#### 2.5.3. Decellularized Extracellular Matrix

Decellularized extracellular matrix (dECM) bioinks with rheological and gel are promising materials for generating functional human tissues [137]. During decellularization, some of the functions of the ECM are preserved, which facilitate the induced differentiation of pluripotent stem cells on dECM scaffolds into tissue-specific cell types [138,139].

In previous studies, various tissue-specific dECMs (fat, muscle, liver, cartilage, and blood vessels, etc.) were successfully fabricated into printable bioinks [134,140,141]. These successful examples enabled skin-derived decellularized extracellular matrix (S-dECM) to be attempted for the formulation of bioinks based on 3D cell printing for skin tissue engineering. Kim et al. [23] used S-dECM bioink to print and fabricate 3D pre-vascularized skin patches for wound healing with infusion of adipose-derived stem cells (ASCs) and endothelial progenitor cells (EPCs). The 3D-printed skin patches loaded with EPCs+ASCs showed a rapid pre-vascularization in vitro within 3 days. Immunofluorescence results after 3 weeks of implantation in mice revealed that wounds with implanted patches exhibited enhanced wound closure, neovascularization, and robust blood flow. In comparison to injection of EPCs + ASCs alone, 3D-printed skin patches using dECM bioink showed significant formation of CD31-positive blood vessels, thus promoting skin repair.

Extracellular matrix associated with cardiac tissues is also widely studied and used in 3D bioprinting today. For example, decellularized cardiac extracellular matrix hydrogel (cECM) is a promising biomaterial for myocardial repair, which can improve cardiac function and progenitor cell delivery [142]. Bejleri et al. [24] fabricated a 3D-bioprinted patch containing cECM for the delivery of pediatric cardiac progenitor cells hCPCs. The patch is printed with bioinks consisting of cECM, hCPCs, and GelMA. Compared to hCPCs grown in pure GelMA patches, the addition of cECM within patch results in a 30-fold increase in cardiogenic gene expression of hCPCs. As seen in the improved endothelial cell tube formation, the media from GelMA-cECM patches revealed an increased angiogenic potential (>2-fold) compared to the GelMA alone group.

Overall, the successful fabrication of vascularized tissue is key to tissue engineering. Still, it is difficult to create a tissue engineering scaffold that is recognized to be very effective in promoting vascularization. Among the many additives added to 3D-printed scaffolds, most of their pro-angiogenic mechanisms are related to vascular growth factors, especially VEGF, such as stimulating VEGF expression in various ways, enhancing VEGF activity, or binding to VEGF to enhance its stability, and so on. Although VEGF plays an irreplaceable role in vasculogenesis and angiogenesis, there are also many other factors or pathways that can contribute to pro-vascularization.

#### 2.5.4. Angiogenic Peptides

Angiogenic peptides (AP) are synthetic peptides that are functionally equivalent to VEGF. AP and VEGF are known to bind to the same receptors to initiate angiogenesis. However, AP can be synthesized on a larger scale and is less expensive than VEGF, thus enabling greater use to promote angiogenesis [143]. Wang et al. [20] made a dual-delivery bone tissue engineering scaffold by low-temperature 3D printing of β-tricalcium phosphate and osteogenic peptide (OP) containing water/PLGA/DCM emulsion—and coating AP on the scaffold surface. After implanting the scaffold into a rat cranial bone defect model for 3 months, the scaffold not only induced an improvement in new bone formation but also a regeneration of blood vessels with larger diameters compared to the control group. The scaffold achieved the dual function of enhancing bone tissue regeneration and vascularization.

## 3. Antibacterial

Human skin and mucosal surfaces are covered by a variety of microbiota, which can be divided into two main groups—resident and transient flora. The resident flora consists of relatively fixed types of microorganisms, while the transient flora consists of non-pathogenic or potentially pathogenic microorganisms [144]. It was experimentally demonstrated that the most common bacterial species is *Staphylococcus aureus* (37%), followed by *Pseudomonas aeruginosa* (17%), *Aspergillus* (10%), and *Escherichia coli* (6%) [145]. The attachment of bacteria to the surface of various substances leads to the formation of biofilms, which allows bacteria to evade the action of antibiotics and causes immune responses [146] and various complications such as the failure of wound healing. In addition, if bacterial membranes adhere to medical implants, they can lead to the failure of the insertion, which can also result in a significant socio-economic burden [147,148]. Furthermore, because any implants will lack blood and lymphatic supply, they have low blood flow and oxygen delivery, making it difficult for leukocytes to exert antimicrobial action. Although many methods were designed to promote vascularization for normal immune surveillance, implants are still at risk of developing infected lesions until an adequate vascular supply is available [149,150]. Therefore, it is imperative to include materials with antimicrobial activity in stents to confer antimicrobial ability to prevent and mitigate infection in implanted stents. Although different substances have different antimicrobial mechanisms, they all show goodness in vivo and in vitro antimicrobial effects. Figure 4 enumerates the different additives incorporated in the hydrogel according to the different antimicrobial mechanisms.

### 3.1. Disruption of Bacterial Cell Membranes

#### 3.1.1. Silver Ion

Silver (Ag) is widely used as an antimicrobial agent to prevent infections during wound healing. Today, silver is mainly used in the form of metallic silver nanoparticles (AgNPs). AgNPs were experimentally demonstrated to have antibacterial activity against bacteria, yeast, and fungi [151,152]. In a study, Ching et al. provided several possible antimicrobial mechanisms: (1) anchoring of AgNps on bacterial membranes leading to rupture of bacterial membranes and leakage of bacterial contents, resulting in bacterial cell death; (2) nanoparticles can penetrate into bacterial membranes and interact with biomolecules, resulting in bacterial dysfunction; (3) AgNps may catalyze the reduction in O_2_ to reactive oxygen species (ROS), leading to downregulation of antioxidant enzyme expression, DNA damage, and apoptosis [153]. Incorporating AgNPs into hydrogel matrices is by physical doping, but it will lead to rapid release of Ag^+^ into the surrounding environment [154,155].

To solve this problem and to exploit the antibacterial ability of Ag^+^, as we can see in Figure 5, Wu et al. [85] designed a superporous polyacrylamide (PAM)/Hydroxypropyl Methyl Cellulose (HPMC) hydrogel with antibacterial ability by using Ag^+^-ethylene interactions and 3D printing technology.

The Ag^+^-ethylene interaction plays an important role in the dispersion, release, and cross-linking of AgNPs in the hydrogel matrix and their cross-linking with the PAM network. Also, this interaction facilitates the cytocompatibility and antimicrobial activity of AgNPs cross-linked hydrogels and reduces the toxicity of Ag^+^. In addition, HPMC was chosen to be incorporated into the formed hydrogels, mainly due to high water absorption and retention properties [156]. It can confer a rapid water absorption rate to the hydrogel to facilitate the retention of more exudate, or extend its use to prevent surface drying of the wound [157]. Antimicrobial experiments in vitro showed that AgNPs cross-linked dressing showed a significant zone of inhibition against both *Staphylococcus aureus* and *Escherichia coli* bacterial colonies compared with the control group of hydrogel dressing containing HPMC and chitosan. Meanwhile, the wounds had smoother surface during healing and minimal scar morphology after 14 days of healing. Histological analysis indicated that the AgNPs cross-linked hydrogel dressing could inhibit the formation of scar tissue.

#### 3.1.2. Photothermal Materials

To improve the antimicrobial properties of stents, photothermal therapy (PTT) attracted extensive interest from researchers as a promising antimicrobial method [158]. Due to its low side effects, deep tissue penetration, and rapid therapeutic effect, it was widely explored for use in the treatment of tumor and implant infections. However, since most photothermal agents do not have the ability to induce tissue regeneration, the development of photothermal agents should promote tissue generation and inhibit bacteria. Nie et al. [25] fabricated a personalized MXene composite hydrogel scaffold GelMA/β-TCP/sodium alginate (Sr2+)/MXene (Ti3C2) (GTAM) with both photothermal antibacterial and osteogenic capabilities using 3D printing technology. As a new class of transition metal carbide/nitride/carbon nitride, it has an ultrathin structure, excellent physicochemical properties, good biocompatibility, excellent near-infrared absorption and photothermal conversion efficiency, low toxicity, and good biodegradability [159,160]. Therefore, MXene was widely used in biomedical applications such as photodynamic therapy (PDT) [161], antibacterial [162], and drug delivery [163], etc. It was noted that MXenes can kill microorganisms such as Gram-positive and negative bacteria by disrupting bacterial membranes directly through physical contact [162,164], especially when supplemented with NIR irradiation for higher antimicrobial efficiency [165]. In vitro antibacterial experiments showed that the GelMA/β-TCP/sodium alginate (Sr^2+^) (GTA) scaffold did not have antibacterial properties, regardless of NIR light irradiation. Although the physical contact of GTAM had certain antibacterial activity, the antibacterial ability was relatively weak, and only the GTAM+NIR group had significant bacterial inhibitory ability. In vivo antibacterial experiments showed that the GTAM scaffold had effective photothermal conversion abilities, and its antibacterial activity was excellent against both G (+) and G (−) bacteria under 808 nm NIR irradiation. On the other hand, GTAM composite bioink was mixed with rat bone marrow mesenchymal stem cells (rBMSCs) and then 3D-bioprinted at low temperature. The scaffolds showed good biocompatibility and promoted the proliferation and differentiation of rBMSCs. The scaffold was implanted into the mandibular defect of rats, and the treatment results under 808 nm NIR irradiation revealed significant bacterial clearance and osteogenesis-promoting effects at the implantation site. In conclusion, this bifunctional 3D-printed composite hydrogel scaffold with both antibacterial ability and the ability to promote precise bone tissue remodeling and regeneration is a novel but very promising biomaterial.

### 3.2. Destruction of Bacterial Cell Walls

#### Vancomycin + Ceftazidime

Vancomycin belongs to a group of glycopeptide antibiotics, which can exert a strong bactericidal ability by inhibiting cell wall biosynthesis through specific binding to the dipeptide d-Ala-d-Ala (AA) at the end of bacterial cell wall precursors [166,167]. Additionally, Ceftazidime can inhibit bacterial peptidoglycan synthesis by inhibiting penicillin-binding proteins, leading to cell wall instability and inhibition of synthesis and cell death [168]. Yu et al. [26] incorporated the combination of two types of antibiotics (vancomycin and ceftazidime) into a 3D-printed scaffold to give the scaffold a sustained release of different antibiotics. They fabricated a mesh-like PCL scaffold by using 3D printing technology and poly(lactic-co-glycolic-acid) (PLGA) nanofibers with a hybrid sheath core structure by using co-axial electrospinning technique. Subsequently, they fabricated this nanofiber membrane in two layers, namely, an electrospun PLGA/vancomycin/ceftazidime layer and a coaxially spun PLGA/BMP-2 layer. After implantation of the scaffold into the femoral defect in rabbits, the drug release profile showed that in vivo release of both antibiotics from the PLGA nanofibers showed a similar trend: the drugs were released explosively on day 1, and the release concentration decreased on day 7 until day 28 when the drug release gradually increased and the release was maintained until day 42. The concentrations of both antibiotics remained well above the 90% minimum inhibitory concentration throughout the study period. BMP-2 also showed a release profile similar to that of the antibiotics. With the stimulation of the scaffold, an abundance of growth factors was induced within the bioactive membrane of the femur. Excellent bone healing ability was found in both radiological examination and biomechanical assessment. Mechanical properties tests showed that femurs in the PCL-PLGA/antibiotic group and the PCL-PLGA/antibiotic/BMP-2 group exhibited torsional strength almost identical to that of healthy femurs. However, the experiment still has some shortcomings: the bone defect model was not an infected bone defect model and the antimicrobial effect was assumed based on the detection of local antibiotic concentrations rather than a direct test of its antimicrobial performance in vivo or in vitro. Furthermore, despite the satisfactory osseointegration, it was unknown whether the antibiotic-loaded nanofibers would behave differently when encountering the healing of large human bone defects that are much more complex.

### 3.3. Inhibits Bacterial DNA Replication

#### Levofloxacin

Levofloxacin, one of the quinolone antibiotics, is a broad-spectrum antibiotic used in the treatment of bone infections to combat and prevent osteomyelitis, penetrate bone trabeculae and bone cortex, and minimize the risk of bacterial resistance [169]. It exerts its antimicrobial action mainly by inhibiting bacterial helicase activity, which leads to the inability of DNA to replicate and synthesize properly, ultimately causing bacterial death. Sadaba et al. [27] produced poly (lactic acid) (PLA) scaffolds with 3D printing technology and, subsequently, added PDA-coated BaSO_4_ particles within the scaffolds and adsorbed levofloxacin to impart antimicrobial properties to the scaffolds. Inorganic particle enhancers such as BaSO_4_ particles can improve the mechanical properties of the polymer and impart additional specific properties to the scaffold [170]. PDA is an adhesive that often acts as a substrate coating to be conjugated with biologically active materials, e.g., small molecule drugs and proteins, to anchor them to the scaffold surface. The PLA/PD-BaSO_4_ scaffold containing levofloxacin effectively inhibited bacterial growth on agar disks inoculated with *Staphylococcus aureus*, and a significant zone of inhibition was observed. Mechanical property analysis of the scaffolds revealed that the composite had simultaneously high stiffness, strength, ductility, and toughness, whereas in non-traditional polymer/inorganic composites, increases in stiffness and strength usually led to significant decreases in ductility and toughness. These advantages of levofloxacin-PLA/PD-BaSO_4_ demonstrate its potential for use in bone reconstruction applications.

### 3.4. Inhibit Bacterial Protein Synthesis

#### Minocycline

Minocycline, which is one of the semi-synthetic tetracycline, has broad-spectrum antimicrobial properties. It exerts its antibacterial effects by inhibiting protein synthesis in bacteria [171]. In addition, minocycline has anti-inflammatory, antioxidant, and anti-apoptotic effects [172]. Compared to common biomaterials, drug delivery platforms incorporating minocycline can promote cell adhesion and proliferation, cytoplasmic diffusion, etc. [172,173]. Martin et al. [28] utilized 3D printing technology to prepare a PLA scaffold and combined the scaffold with collagen, minocycline, and bio-inspired citrate hydroxyapatite nanoparticles (cHA) to finally obtain a multifunctional 3D-printed scaffold. Hydroxyapatite nanoparticles were added to the surface of the scaffold to mimic the composition of natural bone and positively modulate osteoblasts [28,174], so that the multifunctional scaffold structure was closer to the human bone structure. The scaffold has properties such as uniform macropores, adequate wettability, and excellent compressive strength. To evaluate the antibacterial properties of the scaffold, two different antibacterial tests were performed against *S. aureus*. The results of the agar disc diffusion test showed that the scaffold group with the addition of minocycline effectively inhibited the growth of bacteria, and formed a significant zone of bacterial inhibition. In addition, scanning electron microscopy (SEM) evaluated the effect of scaffolds on the activity of *S. aureus* biofilm formation. The surface of PLA-Col scaffolds was completely covered by *S. aureus* biofilm, while no staphylococci were detected on the surface of the scaffold group with the addition of minocycline. In addition, the response of hBMSCs to these scaffolds showed that the addition of cHA significantly stimulated the adhesion, proliferation, and osteogenesis-related gene expression of hMSCs. Overall, the PLA scaffolds obtained by 3D printing successfully combined desirable antibacterial/antibacterial biofilm properties with osteogenic properties. They also maintained morphological and mechanical properties similar to those of human bone, offering the possibility for further clinical trials.

### 3.5. Destruction of Biological Components within Bacteria

#### 3.5.1. NO

NO is a natural molecule that plays an important role in inflammation, cell proliferation, angiogenesis, and wound healing [175]. In addition, NO can play a bactericidal role by damaging DNA, proteins, and lipids of microorganisms. If NO is released in a controlled manner for a long time, it can inhibit bacterial growth on the surface of biological materials [176]. Kabirian et al. [29] designed a 3D-printed small-diameter vascular graft (SDVG) with a NO releasing coating to implant the implant into a CAM. The CAM assay is a cost-effective, simple, and rapid method to assess the biocompatibility and angiogenic capacity of biomaterials and can even be used as a bioreactor [177]. The grafts were printed from polylactic acid and coated with blending of 10 wt% S-nitroso-N-acetyl-D-penicillamine mixed in a polymer matrix consisting of poly (ethylene glycol) (PEG) and PCL to achieve controlled release of NO. The NO release profile showed a burst release of NO on the first day of implantation, followed by a continuous release of NO within the normal physiological range for the next 14 days. The results of in vitro antimicrobial assays showed that the NO-releasing coating inhibited the growth of bacteria and formed a significant inhibition circle. The results of CFU counting method revealed that after 24 h of incubation, microorganisms were reduced by 99.99–100% in the presence of NO-releasing grafts, while bacteria in the control group proliferated stably. In addition, the NO-releasing grafts and control grafts were biocompatible in vitro and in vivo. Compared to controls, NO-releasing SDVGs greatly enhanced the proliferation of ECs and showed significant angiogenic potential in vivo. In conclusion, the NO-releasing 3D-printed SDVGs with accelerated healing and bactericidal properties are a very promising graft.

#### 3.5.2. Natural Plants

The occurrence of antibiotic resistance and multidrug resistance in drug-resistant strains of bacteria reduced the success of traditional synthetic antibiotic therapy. So, a new alternative treatment method derived from natural herbs is an option [178]. *Satureja cuneifolia* (SC) is a natural aromatic plant with anti-Alzheimer’s, anti-diabetic, antioxidant, and anti-microbial properties [179,180]. In addition, it was experimentally demonstrated that SC is rich in phenolic compounds with antimicrobial activity, which are strongly oxidizing and can cause protein coagulation and destroy bacterial proteins thus exerting antimicrobial effects [181]. Ilhan et al. [30] used a 3D printing technique to make SA/PEG composite scaffold and loaded methanol extract of SC into it. The production of 3D-printed scaffold and tissue application of wound dressing are shown in Figure 6. Sodium alginate (SA) in the composite scaffold was all at 9% (*w*/*v*) concentration, while PEG was added at different concentrations of 1, 3, and 5% (*w*/*v*). Phenolic acids and flavonoids in the methanolic extract of SC were analyzed in previous studies. The results show that SC was able to express strong biological activity [182].

The 3D-printed scaffolds showed excellent antimicrobial ability, especially forming a significant inhibition circle around *Staphylococcus aureus* (Gram-positive bacteria), as they contained antimicrobial SC extracts. In addition, the scaffolds exhibited good biocompatibility and low cytotoxicity, and the incorporation of SC generated ideal porosity and mechanical properties of the scaffolds. In conclusion, the SA/PEG/SC 3D scaffold showed great potential for diabetic wound healing and bacterial infection and is expected to be an ideal tissue-engineered wound dressing.

#### 3.5.3. Nanodiamonds

Nanodiamonds (NDS) are increasingly being applied in biomedicine, considering their ability to create a tunable surface [183], and good biocompatibility [184]. Nanodiamond, as an antimicrobial agent, attracted extensive research interest. Regarding its antimicrobial mechanism, the active oxygen-containing groups on the surface of NDS promote their interaction with cellular components to quickly kill Gram-positive and negative bacteria and prevent bacterial adhesion [185]. Nowadays, NDS are mostly incorporated into implants in the form of material coatings or films to function [186,187]. Rifai et al. [31] fabricated selective laser melting titanium (SLM-Ti) scaffolds by using 3D printing and selective laser melting (SLM), and applied nanodiamond (ND) coating on the scaffolds for functionalization modification. As one of the mostly used techniques for manufacturing metal implants, SLM technology often uses powdered metals bonded in a layer-by-layer process to create complex structures, using mainly stainless steel, cobalt-chromium, and titanium. The metals used are mainly stainless steel, cobalt-chromium, and titanium, among others, as they have good biocompatibility and mechanical stability. The concentrations of ND coated on the stent surface varied from 0.075, 0.75, to 7.5% *w*/*v*. The results of in vitro antimicrobial assays showed that after 18 h of growth of *S. aureus* attached to the uncoated and ND-coated stents, the stent group with the lowest ND concentration of 0.075% *w*/*v* had the most *S. aureus* attached, compared to the uncoated control group. After the ND concentration was increased to 0.75% *w*/*v*, the density of *S. aureus* was reduced by 50%, compared to the stent group with ND of 0.075% *w*/*v*. The highest ND concentration (7.5% *w*/*v*) group showed even an 88% reduction in *S. aureus* adhesion. SEM micrographs also confirmed that there was less *S. aureus* adhesion as the ND concentration increased. After inoculating human skin fibroblasts (HDF) and osteoblasts (OB) on four groups of scaffolds and culturing them for 3 days, the maximum HDF and OB cell densities were found on samples at the highest concentration of ND (7.5% *w*/*v*), with a 32% and 29% increase in density, respectively, compared to the control group. Overall, with the use of 3D-printed SLM-Ti in orthopedic implants, modifying its surface with ND provides a simple, rapid, and beneficial cell-implant interface.

### 3.6. Multi-Mechanism Antibacterial

#### 3.6.1. Chitosan

Chitosan (CS) is a natural polysaccharide derived from deacetylation of chitin, and consists mainly of d-glucosamine and N-acetyl-d-glucosamine [188]. Chitosan is now widely used in biomedical applications, including tissue engineering, infection control, and wound healing [188,189]. Its hemostatic and antimicrobial functions can accelerate the transition from the hemostatic to the inflammatory phase after wound trauma and modulate the inflammatory response [190]. Its antimicrobial activity results from the interaction of the positive charge it carries with ions on the surface of negatively charged cells. Such activity disrupts bacterial cell membranes, prevents the transport of cellular material, increases the internal osmotic pressure, and causes the rupture of microbial cells, among other things. In addition, it interacts with bacterial DNA and prevents DNA transcription, thus inhibiting microbial ribonucleic acid (RNA) synthesis [191]. Intini et al. [32] fabricated a 3D chitosan scaffold and implanted it into the skin defects of diabetic rats. Twenty days after implantation, it was shown that the scaffold exhibited good biocompatibility and cytocompatibility. Both MTT assay and neutral red analysis showed that the chitosan matrix was non-cytotoxic. Seeded normal human skin fibroblast (Nhdf) and keratinocyte (HaCaT) cells were viable to grow and colonize the 3D chitosan matrix. An analysis of epidermal wounds in rats revealed that after 7 days, the wounds looked approximately 50% less than the initial area in both animals treated or untreated with chitosan scaffolds, with no significant difference between the two. After 10 days, the wounds treated with chitosan scaffolds had healed, while in the control animals, there was still crusting. After 14 days, the wounds were completely healed in all groups, with only scars visible. Overall, treatment with chitosan scaffolds improved and accelerated wound healing compared to untreated animals, and no signs of infection were observed, probably due to its inherent antimicrobial activity. Histological analysis of the wounds also showed that the chitosan group exhibited more intense tissue reorganization and more mature tissue formation on the wounds.

#### 3.6.2. Zinc Ions

Zinc (Zn) ions, one of the chemical bactericides, have good antimicrobial activity and play an important role in tissue differentiation [192]. The antimicrobial activity of Zn particles was investigated and reported in numerous articles, but mostly in the form of Zn oxide nanoparticles [193,194,195]. Zn oxide nanoparticles are a potential broad-spectrum nano antibiotic to demonstrate their strong antibacterial properties against *Escherichia coli* [196], *Staphylococcus aureus* [197,198], and *Klebsiella pneumoniae* [199], etc. However, its antimicrobial mechanism is complex, and the antimicrobial effect exhibited may be the result of several different mechanisms acting together. For example, Li et al. [200] suggested that due to the negative charge of the bacterial film, an electrostatic reaction between the film and the oppositely charged ZnO nanoparticles leads to the rupture of the film eventually triggering bacterial death. It was also suggested that ZnO nanoparticles can release free Zn^2+^ to disrupt the internal ionic homeostasis of the bacterium and, subsequently, lead to the death of the bacterium. However, there is no unified conclusion on which mechanism is the main one.

Li et al. [81] used 3D printing and bilayer modification techniques to fabricate a 3D ceramic implant with excellent mechanical properties, broad-spectrum antibacterial, and anti-infective properties. They made yttrium oxide-stabilized zirconium oxide (3Y-ZrO_2_) nanopowder into 3Y-ZrO_2_ ceramic by 3D printing, mold plasticity, and sintering. The mechanical properties were enhanced by uniformly coating ISO resin onto the ceramic by a thermal spraying process at high temperature and pressure. Subsequently, ZnO nanosolution was dropped on this ceramic to finally produce ZnO-ISO bilayer modified ceramics [201]. Impact test, bending test, tensile test, and compression test of ISO resin modified ceramics showed that mechanical properties of ISO resin modified ceramics were significantly improved. In addition, the ISO resin did not inhibit the growth of *Staphylococcus aureus* and *Escherichia coli*. There was no significant difference in the number of bacterial colonies between the ISO resin-modified ceramic group and the unmodified ceramic group. However, the ZnO-ISO bilayer modified ceramics showed excellent antibacterial performance against both *S. aureus* and *E. coli*. There was almost no colony formation after treatment of bacteria with ZnO-ISO bilayer modified ceramics. Finally, 14 days after implantation of the bilayer-modified ceramics made by 3D bioprinting, the mice in both the experimental and control groups were normal, with no significant differences in all examinations. This indicates that the ISO resin material has excellent biocompatibility. This study successfully designed a 3D ceramic implant combining excellent mechanical properties and good antimicrobial properties through the bilayer modification technique. But its long-term effects in vivo and in vivo antimicrobial properties are still unknown.

#### 3.6.3. Copper Ions

Copper (Cu) is one of the important elements involved in physiological processes in living organisms and can act as a cofactor for key metabolic enzymes [202,203]. In addition, Cu^2+^ was shown to have efficient antibacterial activity against various bacteria, including *Staphylococcus aureus* and *Escherichia coli* [204]. The mechanism may be that when Cu^2+^ is present as a free ion, it will generate highly toxic hydroxyl radicals from hydrogen peroxide and superoxide [205,206]. The generated hydroxyl radicals can oxidize with most bacterial macromolecules to exert antibacterial effects [207,208]. In addition, Cu^2+^ can be associated with affecting the bacterial outer membrane potential, leading to the rupture of the bacterial membrane, and eventually to bacterial death. Vidakis et al. [83] used 3D printing technology to prepare nanocomposites of medical grade polyamide 12 (PA12) with Cu^2+^ oxide (cuprous oxide) in different ratios (choosing 0.5 wt.%, 1.0 wt.%, 2.0 wt.%, 4.0 wt.%, and 6.0 wt.% weight ratio loading of cuprous oxide). Additionally, 3D printing technology, also known as additive manufacturing (AM) technology, was used in a wide range of these applications [209,210]. Fused filament fabrication (FFF), as one of the AM technologies, is based on thermoplastic polymers and/or composite materials [211,212]. The medical-grade PA12 material used in this experiment is often used as a matrix material, together with metal and oxide fillers in AM applications that require antimicrobial properties in 3D scaffolds [213,214]. In addition, PA12 is commonly used in 3D-printed tissue engineering experiments, as it has excellent toughness properties, the capability to extend extensively before fracture, and the excellent thermodynamic properties [215,216]. As shown in Figure 7 and Figure 8, after culturing the scaffolds in Petri dishes containing Gram-negative *Escherichia coli* and Gram-positive *Staphylococcus aureus* for 24 h, all groups showed good antibacterial activity. The group with a cuprous oxide ratio of 4.0 wt.% showed the highest antibacterial effect against both bacteria.

Other than the above antimicrobial materials and their antimicrobial mechanisms, many other materials can be used as scaffold additives to impart antimicrobial properties to 3D-printed tissue engineering. In conclusion, excellent antimicrobial properties of scaffolds are the prerequisite and foundation for scaffolds to implant and function in vivo, and more antimicrobial materials and different antimicrobial mechanisms can be applied in 3D printed tissue engineering scaffolds.

## 4. Immunomodulation

Immunomodulation refers to the physiological function of the body to recognize and exclude antigenic foreign substances, and maintain its own physiological dynamic balance and relative stability. It is often divided into humoral and cellular immunity, involving various immune cells, such as lymphocytes, macrophages, and NK cells. The immune regulation involved in tissue engineering repair is mostly anti-inflammatory, that is, the regulation of macrophage polarization. Macrophages are one of the main cells that regulate non-specific immune responses, and polarized macrophages play a very important role in inflammatory responses, injury repair, and angiogenesis [217]. Macrophages can be regulated to differentiate into two types—classically activated (M1 type) and alternatively activated (M2 type) [218,219]. M1 type macrophages can be activated by lipopolysaccharide (LPS), T lymphocyte-derived lymphokines, and cytokines involved in T cell-mediated immune responses [220]. After activation, they can secrete pro-inflammatory factors, recruit inflammatory cells against pathogens, and assume many important roles in the early stages of inflammation [221]. However, sustained M1-type activation can cause tissue damage and lead to inflammatory diseases. In contrast, M2-type macrophages are mainly activated by Th2 cytokines such as IL-4, IL-13, and prostaglandin E2, which are responsible for suppressing the inflammatory response and tissue repair effects to restore the tissue to its original state [222,223]. However, excessive activation of the M2 type, in turn, can cause excessive tissue repair and cause fibrosis, among other things. Therefore, finding substances that can balance the polarization of M1-M2 type macrophages is the key to the successful application of tissue engineering [224,225].

### 4.1. Adrenocorticosteroids

#### 4.1.1. Dexamethasone

Dexamethasone (DEX) is a synthetic corticosteroid chemical that is commonly used clinically as an anti-inflammatory agent. It was shown to exert anti-inflammatory effects through various pathways, such as reducing the expression of cyclooxygenase 2 (COX-2). It inhibits prostaglandin production, inflammatory signaling, and neutrophil and macrophage exudation and aggregation to the site of inflammation to exert anti-inflammatory effects [226,227]. Although it was demonstrated that long-term administration of DEX can attenuate the tissue inflammatory response, there are significant side effects such as osteoporosis, gastrointestinal bleeding, and immunosuppression due to long-term systemic exposure to high concentrations of glucocorticoid drugs [228,229]. Therefore, constructing a hybrid scaffold by using 3D printing processing and adding DEX to the scaffold can be a good way to achieve a high dose in the target tissue and a low dose systemically, which can exert anti-inflammatory effects and reduce its side effects at the same time. However, since DEX is a lipophilic substance, its water solubility is low and it is not efficient when loaded into scaffolds [230,231]. Therefore, it is important to utilize a safe medium to load DEX more efficiently into scaffolds. Lee et al. [33] designed a robust and biodegradable 3D tubular scaffold by the combination of electrostatic spinning technique (ELSP) and 3D printing technique, and, subsequently, loaded DEX onto this scaffold using a mild surface modification reaction of PDA, polyethyleneimine (PEI), and carboxymethyl β-cyclodextrin (βCD). Carboxymethyl-βCD is a cyclic oligosaccharide with a hydrophilic outer surface and a central lipophilic lumen [232]. Therefore, carboxymethyl-βCD can form inclusion complexes with DEX, which not only allows massive loading of DEX, but also regenerates tracheal tissue after transplantation and has anti-inflammatory activity [233,234]. To introduce carboxymethyl-βCD, they used PDA and PEI chemical modifications to introduce an amine-rich surface on the PCL scaffold, thus allowing carboxymethyl-βCD to graft through amide bonds [235,236]. Compression tests of this composite scaffold demonstrated that the fabricated 3D tubular scaffold has stronger mechanical properties than natural rabbit tracheal tissue. In addition, atomic force microscopy (AFM) and X-ray photoelectron spectroscopy (XPS) analysis showed that the surface treatment of the scaffold was successful because a large amount of DEX was loaded onto the scaffold [232]. In addition, they quantified the amount of tumor necrosis factor-α (TNF-α) and IL-6 secreted by cells on RAW 264.7 cells before and after LPS treatment by ELISA. As shown in Figure 9, cells on the membrane expressed high amounts of TNF-α and IL-6 under LPS treatment. However, cells on PCLDPβ+DEX expressed significantly lower amounts of TNF-α and IL-6 than cells on PCL, suggesting that the DEX-containing scaffold exhibited significantly enhanced anti-inflammatory activity.

Four weeks after implantation of the stent at the tracheal defect in rabbits, the PCLDPβ stent loaded with DEX exhibited higher cell adhesion and stronger tracheal mucosa regeneration, and formed a more patent airway. In conclusion, this 3D tubular scaffold loaded with DEX has good potential for regeneration of tracheal tissue and anti-inflammatory effects, offering the possibility of future tracheal transplantation therapy.

#### 4.1.2. Prednisolone

As one of the adrenocorticosteroids, prednisolone and DEX have similar anti-inflammatory effects. prednisolone can inhibit the accumulation of inflammatory cells (including macrophages and leukocytes) at sites of inflammation, and their phagocytosis. Furthermore, it can prevent the release of lysosomal enzymes, and the synthesis and release of chemical mediators of inflammation. Farto-Vaamonde et al. [34] utilized two different ways to load prednisolone or DEX into a 3D-printed PLA scaffold. The first one immerses the pre-printed 3D PLA scaffold in a prednisolone solution, which covers its surface with prednisolone and allows for rapid release to exert its antimicrobial properties. The second one immerses the polylactic acid filament in DEX solution to make the polylactic acid swells reversibly. The melting of the polylactic acid during the subsequent 3D printing process helps the effective integration of DEX in the printed filament. It can also achieve continuous slow release for osteoinduction. These two loading methods offer the possibility to generate different drug concentration gradients in the same scaffold and allow different drugs to exhibit different release patterns. The ability of the scaffold to attenuate the inflammatory response was tested by macrophages stimulated with LPS. The blank scaffolds did not attenuate the levels of PEG2 and TNFα, which were similar to those of the positive controls (positive control refers to stimulated macrophage cells). Differently, all scaffolds surface loaded with prednisolone caused a significant decrease in PEG2 and TNFα levels, as effective as free drug dispersed in cell culture medium (ANOVA *p* < 0.001; multiple range test *p* < 0.05). By contrast, scaffolds loaded with DEX did not significantly change the inflammatory response. In addition, all scaffolds showed good biocompatibility with fibroblasts, macrophages, and hMSCs, and loading of various drugs did not impair the printability or mechanical properties of the scaffolds.

### 4.2. Metal Ions

#### 4.2.1. Copper Ions

Copper is one of the essential trace elements and plays an essential regulatory role in the maintenance of physiological homeostasis [237]. In addition to the previously mentioned pro-vascularizing and antibacterial effects, Cu is also important in immune regulation. Many studies demonstrated the positive role of Cu^2+^ in the fight against arthritis, and Cu^2+^ deficiency decreases bone strength and increases the incidence of osteoarthritis [238,239]. The mechanism is that Cu^2+^ plays an important part in regulating the polarization of macrophages. It can inhibit the activation of macrophages and suppress the synthesis and release of inflammatory factors [240]. Lin et al. [84] made Cu-BGC scaffolds by incorporating different concentrations of Cu^2+^ (0.781–25 mg/mL) into bioactive microcrystalline glass using 3D printing technology. After attaching chondrocytes to the scaffold for 12 h, a significant increase occurred on the proliferation of chondrocytes in the Cu-BGC group, as compared to the BGC group and the blank control group. After detecting the HIF pathway using RT-qPCR analysis, the expression level of HIF-1α was significantly enhanced in the Cu-BGC group after 3 days of incubation in ionic extracts, as compared to the other two groups. The relevant genes and HIF-1α genes were subsequently examined in chondrocytes cultured under different concentrations of Cu^2+^. The final results showed that Cu^2+^ ions released from the Cu-BGC scaffold played an important role in promoting chondrocyte proliferation and differentiation and activating the HIF pathway for cartilage repair. Subsequently, the immunomodulatory effect of the scaffold was evaluated by studying the expression of inflammatory cytokines and macrophage surface markers. The results showed that on days 1 and 3, the expression of pro-inflammatory cytokine genes was suppressed and that of anti-inflammatory cytokine genes was increased in macrophages cultured with Cu-BGC ionic extract at a range of concentrations (0.781–25 mg/mL), as compared to the BGC and CTR groups. In addition, the expression of M1 surface marker (iNOS) was significantly suppressed and the expression of M2 surface marker (CD206) was significantly enhanced in macrophages of this group. All these results suggest that the ionic extract of Cu-BGC can promote the shift of macrophages from a pro-inflammatory M1 phenotype to an anti-inflammatory M2 phenotype. Finally, the scaffold was implanted into the cartilage defect on the femoral condyle of rabbits, and in vivo histological studies also showed that the Cu-BGC scaffold group significantly improved cartilage regeneration and enhanced the recovery of the osteochondral interface. Overall, the Cu^2+^, released from this Cu-BGC scaffold, triggered the immune response and suppressed the inflammatory response in osteochondral tissue—by activating the HIF signaling pathway, reducing damage to cartilage tissue, and promoting chondrocyte proliferation and maturation. This may provide a way to prevent and treat osteoarthritis associated with osteochondral defects in the future.

#### 4.2.2. Magnesium Ions

Mg ions are one of the main ions important for maintaining normal physiological functions in humans, and they serves as a cofactor for more than 300 enzymes to play a vital role in regulating the physiological functions of the heart and nerves, and regulate blood pressure and immune regulation [93,241,242]. It was demonstrated in many studies that nuclear factor-κB (NF-κB) is a central regulator of inflammation-induced cytokine production. Also, IκBα shares an activation pathway with NF-κB. Mg^2+^ increase IκBα levels, but lead to reduced NF-κB activation and secretion of cytokines such as interleukin-6 (IL-6) and tumor necrosis factor α [243,244,245]. In addition, it was also found that the Notch1 pathway promotes inflammatory macrophage polarization [246,247], and the interaction of Mg^2+^ and Icariin can synergistically inhibit Notch1 protein expression to exert anti-inflammatory effects. Wang et al. [80] fabricated porous, 3D, Ti_6_Al_4_V scaffolds (PT) and placed them in a 72-well plate. The solutions were prepared: (a) 5 wt.% SF aqueous solution, (b) 1 mg/mL icariin in 5 wt.% SF aqueous solution, (c) 4 mg/mL Mg-MOF-74 in 5 wt.% SF aqueous solution, and (d) 4 mg/mL ICA@MOF in 5 wt.% SF aqueous solution. The four solutions were slowly injected into the PT in the 72-well plate until the PT was completely submerged by the solution. Finally, four groups of samples were achieved: (a) PT/SF (Ti/SF); (b) PT/SF/MOF (Ti/SF/MOF); (c) PT/SF/icariin (Ti/SF/I); (d) PT/SF/ICA@MOF (Ti/SF/MOF/I). Icariin was demonstrated to induce the differentiation of monocytes into macrophages and can function as an inhibitor of the NF-κB signaling pathway through upregulation of the PI3K/Akt pathway, thereby significantly inhibiting the release of pro-inflammatory cytokines such as IL-6 and TNF-α [248,249]. Furthermore, MOFs that are incorporated in scaffolds have excellent drug delivery capabilities due to their excellent chemical properties, biodegradability, and high porosity [250]. MOF-74, one of the MOFs, has low cytotoxicity and controlled drug release properties and can be used as a drug release platform [251]. To further verify the polarization of macrophages, immunofluorescence staining was used to monitor iNOS (green, M1 marker) and Arg-1 (red, M2 marker) in Raw264.7 cultured for 4 days. The results showed that the majority of macrophages in the Ti/SF group were of M1 phenotype (green). M2-type macrophages dominated (red), especially in the Ti/SF/MOF/I group. This was mainly due to the release of a certain concentration of Mg^2+^ in the MOF of this group. Although the concentration of icariin was higher in the Ti/SF/I group, its anti-inflammatory effect was weaker than that of the Ti/SF/MOF/I group, probably because icariin has an optimal concentration and too high concentration may cause cytotoxic side effects and waste of resources, etc. Subsequently, they examined the expression content of Notch1 protein in each group using cellular immunofluorescence and Western blotting. Figure 10 shows that the level of Notch1 protein in the Ti/SF/MOF/I group containing ICA@MOF was significantly lower than the other three groups.

Overall, the Ti/SF/MOF/I group had the highest M2 macrophage intensity and the lowest Notch1 protein expression level, which was mainly due to the synergistic immunomodulatory effects of Mg^2+^ and icariin released continuously from SF/ICA@MOF. In addition, they implanted the composite PT scaffold into the distal femoral medulla of osteoporotic rats and showed abundant new bone formation at both the peripheral and internal sites of the implantation site. The scaffold had good mechanical properties and biocompatibility, which significantly improved bone metabolism and promoted osseointegration. The scaffold greatly exerted anti-inflammatory effects and significantly improved osteoporotic integration using the sustained controlled release of icariin and Mg^2+^ from MOFs, offering the possibility of clinical practice for titanium prostheses in patients with osteoporosis.

### 4.3. Animal Sources

#### 4.3.1. Interleukin-4

Interleukin-4 (IL-4) is one of the cytokines of the T helper (Th2) family of cells [252]. The activation of the typical pathway of IL-4 production causes phosphorylation of STAT6 and upregulation of GATA-binding protein 3 (GATA3) expression [253], which, in turn, constitutes a major factor in the transcriptional regulation of Th2 cells. Therefore, IL-4 can activate the production of Th2 cytokine and, thus, promotes the polarization of M2-type macrophages. In addition, IL-4 has the ability to antagonize the Th1-driven pro-inflammatory immune response, as evidenced by the downregulation of the synthesis of pro-inflammatory cytokines such as IL-10 and TNF-α [254] and the inhibition of pro-inflammatory chemokines [255]. Therefore, IL-4 was used in 3D-printed tissue engineering due to its excellent immunomodulatory ability. Wang et al. [35] used GelMA-Dextran (PGelDex) as bioink and incorporated both IL-4-loaded Ag-coated gold nanorods (AgGNRs) and hMSCs to confer anti-infective and immunomodulatory functions to the scaffold. Subsequently, to verify the immunomodulatory efficacy of IL-4, they performed an inflammation-inducing treatment of the bioink with IFN-γ and LPS. After 24 h of incubation, qRT-PCR analysis showed a significant decrease in M1 macrophage markers and a significant increase in M2 macrophage markers in the IL-4 and IL-4+MSC groups, as compared to the control group. The results suggested that bioink loaded with IL-4 and MSCs successfully polarized M1 macrophages to M2 type and exerted anti-inflammatory effects. There were many studies demonstrating the possible therapeutic inflammatory and immunomodulatory functions of MSCs [256], but their specific intrinsic molecular pathways and mechanisms of action are still under further investigation. Overall, they loaded two functional agents, namely IL-4@AgGNRs and MSCs, in the hydrogel. AgGNRs have the ability to eliminate the infection. Additionally, the synergistic effect of IL-4 and MSCs could induce the polarization of macrophage phenotype from M1 to M2 phenotype. Ultimately, a composite scaffold was made with simultaneous antibacterial, anti-inflammatory, and cytoarchitectural properties.

#### 4.3.2. Mesenchymal Stem Cells

The function of MSCs in the immune regulation in tissue repair was demonstrated in various disease models [257,258]. However, their specific intrinsic anti-inflammatory molecular mechanisms are still under investigation due to their complex and diverse cell composition. It was suggested that it can act as a modulator of the inflammatory response, with paracrine activity being markedly promoted during inflammation [257]. It can result in the secretion of large amounts of cytokines and factors that induce a shift from an M1-type pro-inflammatory phenotype to an M2-type anti-inflammatory phenotype in macrophages [259], and a reduction in the production of pro-inflammatory cytokines by macrophages [260]. Paul et al. [122] used hydrogel-coated endometrial mesenchymal stem cells (eMSCs) and 3D melt electrospun wire nets to generate tissue-engineered scaffolds as a potential therapy for pelvic organ prolapse (POP). They firstly fabricated 3D-printed PCL mesh by using melt electrospinning (MES) at different temperatures, followed by optimizing the aloe vera-SA (AV-ALG) composite hydrogel ratio to 1:1 (1% AV-1% ALG), and finally bioprinted purified eMSCs in this composite hydrogel onto the MES-printed mesh. Implantation of the scaffold into the subcutaneous tissue of NSG mice for 1 week revealed that eMSCs printed onto MES promoted tissue integration and eMSCs retention. Detection of macrophage phenotype at the implantation site revealed a significant reduction in CCR7M1-type macrophages within the MES-Hyd group and MES-Hyd-eMSCs group, as compared to the control MES group. This suggests that AV-ALG hydrogel inhibits macrophage responses and functions to reduce inflammation compared to the MES control group. Furthermore, the addition of eMSCs further reduced the M1 inflammatory macrophage response and increased the M2 macrophages, as compared to the MES-Hyd group. Macrophage quantification showed significantly higher total F4/80+ macrophages in the control MES group and the MES-Hyd-eMSCs group compared with MES-Hyd, where the majority of the MES-only group were M1-type macrophages, whereas the majority of the F4/80 macrophages in the MES-Hyd-eMSCs group were anti-inflammatory M2-type macrophages of the CD206 phenotype. Overall, the addition of eMSCs to the scaffold exerted a regulatory role on the inflammatory response and provided a suitable candidate for tissue engineering repair of POP.

#### 4.3.3. Ac2-26 Peptide

Ac2-26 peptide, the peptide located at the N-terminal end of ANXA1 of 26 amino acids, was found to inhibit TNF-α production in monocytes, inhibit NF-κB signaling of the proinflammatory pathway [261], and promote phagocytosis of neutrophils [262,263]. Xu et al. [36] fabricated a polylactic acid/4-arm polyethylene glycol hydrogel (PCL@tetra-PEG) composite scaffold as a tissue engineered meniscus (TEM). The PCL scaffold was fabricated with tapering porous and heterogeneous structures, and BMSCs were then seeded on them. The PCL@tetra-PEG scaffold was subsequently generated in a metal mold using a rotation-immersion method, in which tetra-PEG hydrogel not only improves the biomechanical properties of the PCL@tetra-PEG scaffold to fit the native meniscus, but also helps achieve different regional encapsulation and spatiotemporal zonal release of two human growth factors (CTGF and TGF-β3). Significantly, the simultaneous encapsulation of Ac2-26 peptide within the composite scaffold can perform anti-inflammatory and antioxidant effects, regulate the complex microenvironment, and promote tissue regeneration.

In vitro inflammatory cell model experiments showed that high concentrations of Ac2-26 peptide could effectively inhibit the production of ROS, and reactive nitrogen species (RNS) in macrophages which were induced by LPS. The Western Blot results showed that the expression of M1-type macrophage markers iNOS and COX-2 were significantly increased after LPS treatment. In contrast, after 12 h of Ac2-26 peptide treatment, the expression of iNOS and COX-2 decreased significantly as compared with the control group, and the higher the concentration of Ac2-26 peptide was, the more obvious the down-regulation of iNOS expression was. Moreover, in contrast to M1 markers, M2-type markers including Arg-1 and IL-10 were significantly upregulated after Ac2-26 peptide treatment. All these results indicated that Ac2-26 peptide promoted the phenotypic transformation of macrophage phenotype from M1 to M2 type, and exerted anti-inflammatory and antioxidant effects, to favor the promotion of tissue regeneration. Moreover, they used immunofluorescence and qPCR to assess the role of different types of pores on inducing tissue differentiation at the defect. It showed that the progressive pore meniscal scaffold was more favorable for inducing fibrocartilage differentiation of MSCs, and could induce in vitro expression of COL-1 and COL-2 with more heterogeneity than that of the homogeneous pore meniscal scaffold.

### 4.4. Plant Sources

#### 4.4.1. Curcumin

Curcumin, a hydrophobic polyphenol, is the main component of turmeric. Demethoxycurcumin and bis-demethoxycurcumin can also be extracted from turmeric, but they both have lower activity levels than curcumin [264,265]. Curcumin was demonstrated to have the ability to target and regulate several molecules, including inflammatory cytokines, growth factors, and apoptosis-related proteins. It has multiple functions such as anti-inflammatory, antioxidant, and antitumor effects. The mechanism of its anti-inflammatory effects was investigated by its inhibitory effect on cytokine production and expression [266,267]. It was found that curcumin can exert anti-inflammatory activity in LPS/interferon-γ (IFN-γ)-treated macrophages through several mechanisms, for example, by blocking (NF-κB and signal transducer and activator of transcription 1 (STAT1) signaling pathways, thereby inhibiting LPS-induced IL-6 expression in RAW264.7 cells [268]. In addition, flavin derivatives can also inhibit NO, TNF-α, and IL-1β expression by suppressing the mitogen-activated protein kinase (MAPK)/NF-κB pathway in IFN-γ/LPS-stimulated macrophages [269]. Chen et al. [37] made mesoporous CS (MesoCS/curcumin) scaffolds with curcumin. MesoCS nanoparticles were first prepared using a template [270], followed by dissolving turmeric as a stock solution in 0.5 M NaOH to a concentration of 0 mg/mL, 2 mg/mL, 5 mg/mL, and 10 mM (i.e., C0, C2, C5, and C10, respectively), and finally, mixing the scaffolds at the same 0.4 mL/g liquid/powder ratio to make the scaffolds. The hMSCs were cultured in this composite scaffold and the expression levels of TNF-α and IL-1 were detected after LPS treatment for 1, 3, and 7 days. The results showed that the expression levels of TNF-α and IL-1 were significantly inhibited in the C2 and C5 groups, as compared with the C0 group (*p* < 0.05). The expression levels of TNF-α were not significantly decreased in the C10 group. This suggests that scaffolds containing certain concentrations of curcumin have significant anti-inflammatory potential to inhibit LPS-induced TNF-α and IL-1 expression. In contrast, high doses of curcumin (C10) may cause apoptosis mediated by metal ions (e.g., Ca ions) [271,272]. Furthermore, the MesoCS/curcumin scaffold showed good biocompatibility and physical properties, and the addition of curcumin did not alter the crystalline properties of MesoCS. Overall, the MesoCS/curcumin scaffold prepared in this study is expected to be a new material for bone tissue engineering and bone regenerative medicine. Although the side effects of curcumin such as hemodilution [273] and decreased blood glucose [274] remain to be addressed, the scaffold will have great potential for further clinical practice.

#### 4.4.2. Isoflavones

Isoflavones are among the most common phytohormones and soybean and its products are the largest dietary source of them [275]. Genistein, one of the classes of isoflavones, has a wide range of biological activities, such as inhibition of cancer growth [276], promotion of osteoblast and alkaline phosphatase activity [277], and immunomodulation [278,279]. The excellent immunomodulatory effects of genistein are attributed to its multiple anti-inflammatory mechanisms, such as its ability to eliminate free radicals produced by macrophages during inflammation, thereby reducing the release of various inflammatory mediators induced by ROS [280,281]. It has the ability to inhibit the phosphorylation of IκB, an inhibitor of NF-κB, thereby reducing the degradation of IκB, inhibiting NF-κB activation, and, subsequently, reducing the pro-inflammatory cytokine production [282,283] among others. Sarkar et al. [38] loaded three soy isoflavones (Genistein, Daidzein, and Glycitein) in a 5:4:1 ratio, mimicking their original ratios in soy, on 3D-printed TCP scaffolds with pores. In vitro release profile showed controlled release in acidic and physiological buffered media over 16 days. Results of osteosarcoma cell culture experiments showed a 90% reduction in osteosarcoma (MG-63) cell viability and proliferation capacity after 11 days of cultivation, which implies the potential preventive ability of the scaffold against osteosarcoma cell lines. In addition, the scaffold showed good biocompatibility: the scaffold exhibited the ability to promote osteoblast proliferation and differentiation under both static and dynamic conditions. Finally, the scaffold was examined for in vivo immunomodulatory capacity. Twenty-four hours after implantation of the scaffold into the defective distal femur of rats, H&E staining examination revealed reduced neutrophil recruitment at the implantation site. Overall, they conferred in vitro chemopreventive, osteoclast-promoting, and immunomodulatory abilities to the scaffold by incorporating isoflavones and designing a versatile bone graft substitute. The downside is that the anti-inflammatory properties of this experiment were not analyzed for macrophages, polarization, and inflammatory factor concentrations. Long-term in vivo studies are needed.

#### 4.4.3. Quercetin

Quercetin is a flavonoid monomeric compound found in natural plants with various physiological effects such as anti-inflammatory, antioxidant, antitumor, and osteogenic [284,285]. Quercetin was found to impact the M1/M2 phenotype transition in macrophages, inhibiting M1 phenotype polarization and promoting its polarization to the M2 phenotype [286]. Because it can induce phosphorylation of protein kinase B and promote signal transduction and nuclear translocation of transcriptional activator protein 6 (STAT6), it can increase the expression of the M2 phenotype upon binding to anti-inflammatory cytokine-related genes [287]. In addition, there were experiments that demonstrated that quercetin can block the NF-κB signaling pathway, thus promoting M2 polarization for anti-inflammatory repair effects [288]. Wang et al. [39] fabricated layered micro/nano surfaces on the surface of Ti_6_Al_4_V implants using 3D printing technique, alkali heat treatment, and hydrothermal treatment after deposition of titanium dioxide (TiO_2_) on this surface. This increased the adsorption of specific substances on Ti_6_Al_4_V implants and made them more biocompatible. Since quercetin has an excellent ability to chelate metal cations [289], it can be adsorbed as a monomer on TiO_2_, effectively creating a quercetin coating on the surface of the 3D-printed Ti_6_Al_4_V implant. SEM results of 24-h adhesion of RAW 264.7 cells in vitro showed that macrophages on the quercetin-containing coating group were mostly polarized to the M2 phenotype and had more pseudopods. This indicated there were more adequate attachments and diffusions. RBMSCs proliferation also showed similar results. Representative cytokine concentrations secreted by macrophages with M1/M2 phenotypes were further examined by ELISA. The M1 phenotype secreted reduced concentrations of IL-1β cytokines, whereas the M2 phenotype secreted increased concentrations of VEGF-α cytokines. Furthermore, after 7 days of cell adhesion, the addition of quercetin coating elevated the ALP expression on the surface of the motifs, as compared with the control group, thus revealing that quercetin does have the function of promoting osteogenic differentiation of rbMSCs. Finally, the scaffold was implanted into the femoral condylar defect in rats. Two weeks later, the control group showed a severe inflammatory response compared with the group loaded with quercetin. A large quantity of inflammatory cells such as monocytes, neutrophils, and macrophages, and a significant expression of the M1 macrophage biomarker IL-1β, was found in the implantation site. In contrast, inflammatory cells were significantly reduced, and IL-1β expression was significantly decreased in the loaded quercetin scaffold. These indicated the superior anti-inflammatory effect of quercetin at the site of inflammation in vivo. Overall, the loading of quercetin conferred osteointegration-promoting and immunomodulatory properties to the 3D-printed titanium implants, offering the possibility of future applications in clinical osteointegration.

## 5. Regulation of Collagen Deposition

Collagen, the most abundant protein in mammals (25–30% of total protein mass), is one of the main components of connective tissue. It is closely related to disease and aging, cell proliferation, migration and differentiation, and fibrosis formation [290]. More than 20 types of collagen were identified, of which the most abundant and well-studied are type I, type II, and type III collagens [291]. In addition, collagen is mostly found in the interstitial matrix of thin-walled organs and serves as a scaffolding structure to stabilize and maintain tissue integrity. Therefore, imbalance of collagen metabolism plays a crucial role in the development of certain diseases. In the middle and late stages of injury repair, however, an imbalance in the ratio of type I to type III collagen (type I collagen replaces type III collagen), and often leads to fibrotic lesions and scar formation [292,293]. Collagen metabolism is regulated by signaling pathways such as TGF-β/Smad, PI3K/Akt, MAPK, and NF-κB, and each pathway interacts with each other to mediate collagen metabolism [294,295,296]. Therefore, there are many studies that regulate collagen deposition by adding some kind of signaling pathway modulating substance to 3D-printed tissue engineering scaffolds or adding some kind of collagen-containing substance directly to the scaffold to make the collagen deposition at the scaffold implantation injury repair with specific tissue specificity. Thus, they can better promote efficient and scar-free repair at the implantation site and promote tissue regeneration.

### 5.1. Direct Regulation

#### 5.1.1. Collagen

As mentioned above, different types and ratios of collagen deposition play different roles in different periods of wound repair. So, it is crucial to form tissue-specific collagen deposition at the implantation injury site, to promote regeneration of wound repair tissue, and to avoid scar tissue formation due to excessive collagen deposition. Martin et al. [28] used 3D printing technology to prepare a PLA-collagen -minocycline-nanohydroxyapatite scaffold; the specific preparation process was mentioned in the Section 3. They added collagen directly to the 3D-printed scaffold. The scaffold had good osteogenic properties, morphological and mechanical properties matching those of bone trabeculae. The addition of collagen enhanced the structural strength of the scaffold and formed a scaffold support closer to the scaffold itself. It also had a positive effect on bone differentiation and metabolism.

#### 5.1.2. Extracellular Matrix

In spite of the good effects from direct addition of collagen to scaffolds, rapid degradation and shrinkage are inevitable after long-term implantation, and so, their lifespan is very limited [297,298]. The application of dECM as an ink for 3D bioprinting can be a good solution. Extracellular matrix is mainly composed of various components such as collagen, non-collagen, and elastin [299], and collagen as the main component can not only give the extracellular matrix the function of directly regulating collagen deposition, but also avoid the degradation and shrinkage process caused by a single collagen. Kim et al. [23] used S-dECM bioink to print and fabricate 3D pre-vascularized skin patches for wound healing with infusion of ASCs and EPCs. After implanting the patches into mice for 2 weeks, the 3D cell-printed skin tissues using dECM as bioink were very stable, had less shrinkage, and contained abundant type I and III collagen, as well as fibronectin with better cellular integrity as compared to the single collagen group. In addition, the cytoskeleton of tissue repair and epithelialization (KGF-1)-related gene expression levels were also higher in the dECM-as-bioink group. Therefore, using extracellular matrix as a 3D bioprinting ink could better regulate collagen deposition and promote tissue repair and regeneration. However, the issue of cell-related immunogenicity of its heterologous tissues should not be ignored, and the cells of donor tissues need to be completely removed, and so, the use of dECM as bioink for preclinical or even clinical applications still needs to be carefully experimented and considered before use.

### 5.2. Indirect Regulation

#### 5.2.1. Copper Ion

Cu^2+^ is one of the essential ions for the human body. As mentioned above, Cu ions have antibacterial, immunomodulatory, and pro-vascularizing effects, and play a huge role in human metabolic processes. It was demonstrated that Cu^2+^ ions can upregulate HIF-1 expression by activating and binding to the α-subunit of HIF-1 factor, and inhibiting the activity of HIF-1 and the inhibitory factor of HIF-1 [104,300]. The activation of the HIF pathway further stimulates the expression of SOX-9, which leads to increased levels of COL II and ACAN expression in chondrocytes increased [301,302]. This regulates collagen deposition and promotes chondrocyte synthesis and development. Lin et al. [84] fabricated Cu-BGC scaffolds by doping different concentrations of Cu^2+^ (0.781–25 mg/mL) into bioactive microcrystalline glass with 3D printing technology. The results showed that the expression of SOX9 in cartilage was significantly increased after treatment with Cu^2+^. In addition, the expression of COL II and ACAN genes in chondrocytes was also significantly elevated after day 3 of Cu^2+^ treatment. In vitro results also showed that Cu^2+^ concentration in the range of 0.5–16 ppm significantly increased the expression of COL II and NCAD proteins. Chondrocyte proliferation and maturation were stimulated and elevated compared to the control group.

#### 5.2.2. Silk Fiber

In addition to the above-mentioned pro-vascularization functions, SF was also shown to be able to regulate collagen deposition to some extent. As one of the filamentous proteins, *Antheraea assama* SF can improve cell fate and promote ECM secretion owing to its Arg-Gly-Asp (RGD) content. Bandyopadhyay et al. [40] mixed a blend of SF from mulberry (*Bombyx mori*) and non-mulberry (*Antheraea assama*) silk with gelatin as inks for 3D bioprinting to simulate the extracellular microenvironment of the meniscus. Subsequently, EDC and NHS were chemically cross-linked to enhance their stability and mechanical strength. Finally, porcine meniscal fibrocartilage cells were seeded in the scaffold. After 7 days, the expression of sox-9, aggrecan, and collagen II genes were significantly increased. After 21 days, a significant increase in the secretion of sGAG and the deposition of ECM components (i.e., collagen) was detected. This suggests that *Antheraea assama* SF can indirectly regulate collagen deposition by promoting the secretion of ECM for implant site-specific collagen deposition.

#### 5.2.3. Curcumin

Curcumin is one of the main active components of the rhizome of the turmeric plant. In addition to the above-mentioned functions of promoting angiogenesis, immunomodulation, and antioxidation, Bose et al. [41] also demonstrated that curcumin has the function of inhibiting the NF-κβ signaling pathway, thus promoting osteoblast differentiation and the secretion of ECM. They fabricated a PCL-PEG+curcumin-coated TCP scaffold using 3D printing technology. After implanting the scaffold into rats for 6 weeks, this scaffold group showed significant new bone formation and mineralization, as compared with the control group. Tissue staining showed various types of type II collagen, and the scaffold implantation greatly promoted ECM formation and angiogenesis. The results suggest that curcumin promotes the formation and deposition of ECM by promoting osteogenesis, thereby indirectly and specifically regulating collagen deposition.

#### 5.2.4. Mesenchymal Stem Cells

MSCs are multipotent differentiated cells that can be induced to differentiate into stromal cells, osteoblasts, hepatocytes, and other cells and were widely used in disease models to regulate wound healing and tissue regeneration [303,304]. It was demonstrated that MSCs play a major role in regulating collagen production and inhibiting scar formation during the proliferative and mature phases of injury repair.

The main mechanisms of MSCs for scar inhibition are as follows. 1. Inhibition of inflammation in scar tissue. In detail, excessive inflammatory response is one of the main causes of scar formation. MSCs can shorten the inflammatory period and reduce the inflammatory response by acting as a chemotactic agent for various cytokines that contribute to wound healing, such as TNF, IL-8, PDGF [305]. In the meanwhile, they can immunomodulate inflammatory cells in scar formation by secreting various factors and extracellular vesicles [306]. 2. MSCs can affect fibroblast activity in scar formation and can reduce scar formation by inhibiting the gene expression of TGF-β1 and decreasing type I collagen production [307]. In addition, MSCs can inhibit fibroblast proliferation and migration, as well as inhibit angiogenesis in scar tissue. 3. MSCs can play a positive role in collagen remodeling in scar tissue, and can regulate the expression of matrix metalloproteinase (MMP) by secreting tissue inhibitor metalloproteinase (TIMP), thereby delaying or reducing scar formation [308]. In details, in a rat skin burn model, MSCs not only promoted the proliferation of epithelial cells at the skin wound, but also increased the expression of CK19, PCNA, Col I, and Col III [309]. Paul et al. [122] used hydrogel-coated eMSCs and 3D melt electrospun wire nets to generate tissue-engineered scaffolds for POP. The specific production process was mentioned in the previous section. Picro Sirius red staining in Figure 11 shows collagen deposition in and around the reticular area of the MES-Hyd-eMSCs group as compared to MES and MES-Hyd alone. MES-Hyd shows no collagen within the first week of implantation.

The fate of the mesh after 1 week of implantation in vivo was assessed by SEM in Figure 12, showing little degradation of the scaffold after 1 week, as expected for a PCL-based mesh. The deposition of new collagen fibers appeared lower in MES alone (Figure 12A–C) and MES_Hyd (Figure 12D–F). On the other hand, MES_Hyd_eMSCs showed better evidence of tissue integration and new collagen deposition (Figure 12G–I).

All these results suggest that the addition of MSCs to scaffolds can better regulate collagen deposition and promote tissue integration and regeneration. Additionally, among the many different sources of MSCs, human gingival mesenchymal stem cells (hGMSCs) are abundant and readily available [310,311]. In addition, gingival cells have a mechanism similar to fetal tissue healing, and exhibit greater ability to heal scar-free wounds than dermal fibroblasts [312]. Shafiee et al. [313] fabricated a bionic medical polycaprolactone (mPCL) dressing with hGMSCs using 3D printing technology. The medical polycaprolactone dressing has a pore structure and excellent mechanical properties that mimic the layered structure of skin collagen fibers. This facilitates skin wound healing and reduces scar formation. After implantation of the dressing containing hGMSCs into rats for 6 weeks, wound contraction was significantly reduced in the mPCL dressing group, as compared with the control group. Tissue regeneration was significantly improved by granulation and re-epithelialization. In addition, Masson staining showed significant collagen I deposition in the wound area, and the collagen I deposition exhibited an ordered, directionally dispersed narrower form. Overall, mPCL dressings containing hGMSCs resulted in more rapid wound healing, regulated collagen deposition, and reduced scar formation in the rat model.

#### 5.2.5. Granulocyte Colony-Stimulating Factor

Granulocyte colony-stimulating factor (G-CSF) is a hematopoietic factor produced by lymphocytes, macrophages, and endothelial cells, and it promotes the proliferation and differentiation of bone marrow hematopoietic cells [314]. In addition, as an anti-inflammatory protein, endometrial stem cell-derived G-CSF can reduce scar formation by reducing Gli2 protein expression levels to regulate collagen deposition and reduce endometrial fibrosis [315,316]. However, G-CSF has a short half-life in vivo due to receptor-mediated clearance, renal metabolic elimination, and enzymatic degradation [317,318]. Furthermore, due to the continuous migration of G-CSF in the uterine cavity, it cannot maintain a consistently high concentration at the endometrial injury to achieve good therapeutic efficacy. Therefore, to maintain the long-term drug concentration of G-CSF at the site of endometrial injury and to reduce the adverse consequences of a single high-dose administration, it is of great significance to construct a sustained release system of G-CSF. Wen et al. [42] made G-CSF-loaded slow-release microsphere (G-CSF-SRM) hydrogel scaffolds with 3D printing technology. They mixed recombinant human G-CSF, dextrose, and polyethylene glycol in a mass ratio of 1:5:50, and stirred and freeze-dried them. Then, the obtained G-CSF-dextran-PLGA microspheres were processed by curing, hardening, and freeze-drying to complete the G-CSF microsphere preparation. Finally, the prepared G-CSF microspheres were mixed and printed with 20% gelatin and 4% SA to make a 3D-printed G-CSF-SRM hydrogel scaffold. The in vitro cumulative release of G-CSF over time was measured by ELISA. It showed that G-CSF was continuously released from the microspheres for more than 1 month, and the drug release became slow and regular after the explosive release just after implantation. Five weeks after implantation of the scaffold at the endometrial injury in rats, HE staining results showed that the number of endometrial cells and good endometrial regeneration were significantly increased in the group implanted with 3D-printed G-CSF-SRM hydrogel scaffold, whose endometrial thickness and number of glands were significantly higher than the remaining four groups. However, their inflammatory cells and inflammatory factors IL-1β, IL-6, TNF-α expression levels were lower. In addition, Masson staining results showed that this group had relatively less collagen deposition and less fibrosis. Thus, endometrial inflammatory cell infiltration and endometrial fibrosis were significantly reduced due to G-CSF, which effectively prevented fibrous exudation-mediated adhesion and scar formation.

#### 5.2.6. Piezoelectric Effect

It was demonstrated that bioelectricity plays a crucial role in promoting wound healing [319]. However, the gradual weakening of bioelectric stimulation with wound healing may cause disruption of gene regulation, leading to downregulation of the wound healing cascade and eventually disorganized deposition of collagen fibers and abnormal remodeling of the ECM, which delayed wound healing and resulted in the formation of scar tissue [320,321]. Therefore, to solve these problems, Liang et al. [43] fabricated a novel ZnO nanoparticle-modified PVDF/SA piezoelectric hydrogel 3D scaffold (ZPFSA). As one of the commonly used piezoelectric materials, polyvinylidene fluoride (PVDF) has excellent piezoelectric properties, biocompatibility, thermal stability, and resistance to chemical irritation [322]. However, since PVDF is a superhydrophobic polymer, it is contrary to the wetting behaviors of hydrogels and preparing them as hydrogels is a great challenge [323]. Therefore, the incorporation of SA precisely reduces the hydrophobicity of PVDF and creates the possibility of preparing hydrogels from its 3D printing. Moreover, the addition of nanoparticles can enhance the piezoelectric properties of PVDF by promoting the polarization of PVDF [324]. The 3D-printed ZPFSA scaffold was implanted into the defective skin on the back of rats, and the immunohistochemical results showed that the CD31 expression in the ZPFSA 0.5 group was higher than the other groups on day 7 after implantation. This indicated that the wound tissue in the ZPFSA 0.5 group had active angiogenesis to ensure adequate blood supply to the wound. α-SMA is an α-smooth muscle actin secreted by myofibroblasts, which has high expression in the middle and late stages of tissue healing and proliferation to promote wound contraction. However, at the end of wound healing, myofibroblast apoptosis causes a gradual decrease in α-SMA expression to prevent excessive contraction of collagen fibers and formation of scar tissue [325]. At day 14 after implantation, the wound in the ZPFSA0.5 group was close to healing and its α-SMA content was maintained at a low level. The other groups were still in the late stage of healing and had higher α-SMA content. Thus, the ZPFSA0.5 group reduced the possibility of scar formation by accelerating wound healing. In addition, immunohistochemical results showed that the expression of three growth factors, EGF, VEGF, and TGF-β, was higher in the ZPFSA 0.5 group than in the other groups, implying that the electrical stimulation was generated by the ZPFSA piezoelectric scaffold when receiving stress. This could promote the production of cellular growth factors, which greatly accelerated the wound healing process and effectively prevented the formation of scar tissue.

## 6. Future Perspective and Concluding Remarks

This review outlined current important advances in 3D bioprinting of tissue and/or organ engineering. The different additives were classified into four categories according to the different functions they conferred on the scaffold: vascularization, antibacterial, immunomodulation, and regulation of collagen deposition. The mechanisms and the methods of incorporating bioinks were detailed. Many additives can fulfill two or more functions from their excellent properties. For example, many metal ions can not only be antibacterial, but also perform immunomodulatory functions. Extracellular matrix can not only promote vascularization, but also regulate collagen deposition and reduce scar tissue formation, etc. All these summaries offer the possibility of adding small amounts of additives to achieve multiple effects in the future, which will facilitate 3D-bioprinted scaffolds to perform their functions and promote tissue regeneration more efficiently and effortlessly. However, although many experiments in animals or in vitro achieved good outcomes, only a few printed scaffolds were applied to human experiments for research. We think there are many reasons for this; for example: these constructs need to be specific to the relevant characteristics of the graft site (appropriate cell type, ideal mechanical properties, and other required associated cues, such as neovascularization, low inflammatory properties, sterility, and appropriate collagen deposition), which is often a challenge for transplantation [326]. So, future work can be dedicated to the suitable scaffolds to general clinics. Suitable additives can be used according to the different specific environments of the implantation site, thus achieving specific functional modulation, or even meeting one additive to achieve multiple functions, greatly improving transplantation success, reducing the complexity and economic cost of bioprinting, and, finally, facilitating the translation of the constructs from animal experiments to the clinic. Looking forward to the future, it is bright to design the tissue-specific 3D-bioprinted scaffolds and add efficient additives to achieve multi-functions, and then use them in tissue regeneration in human treatment.

## Figures and Tables

**Figure 1 pharmaceutics-15-01700-f001:**
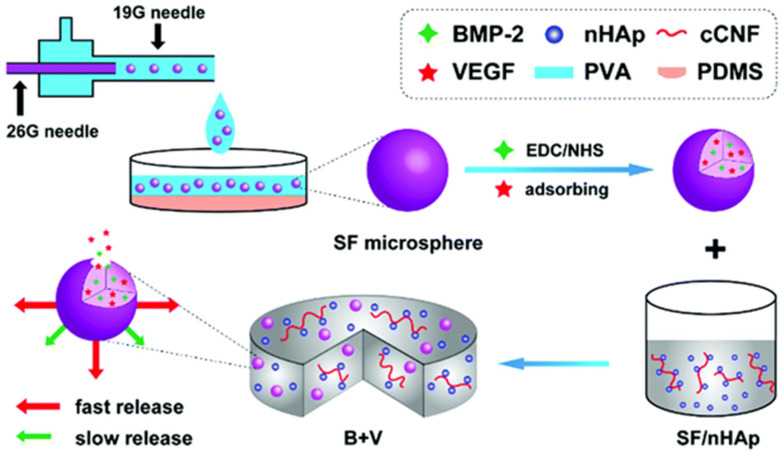
Schematic of the preparation of the B + V scaffold [16].

**Figure 2 pharmaceutics-15-01700-f002:**
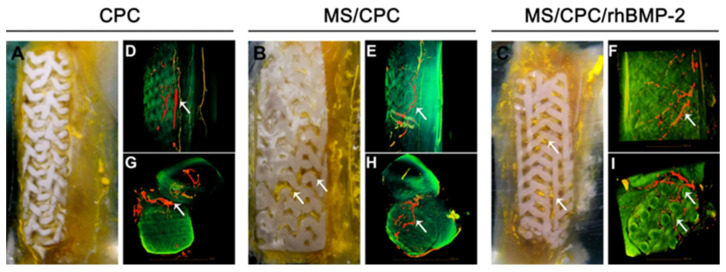
(**A**–**C**): Digital camera photographs of PMMA-embedded blocks from longitudinal sections and 3D reconstructed μCT images of blood vessels from (**D**–**F**) side view and (**G**–**I**) top view of CPC, MS/CPC, and MS/CPC/rhBMP-2 scaffolds after 4 weeks of implantation. White arrow: newly formed blood vessels [77].

**Figure 3 pharmaceutics-15-01700-f003:**
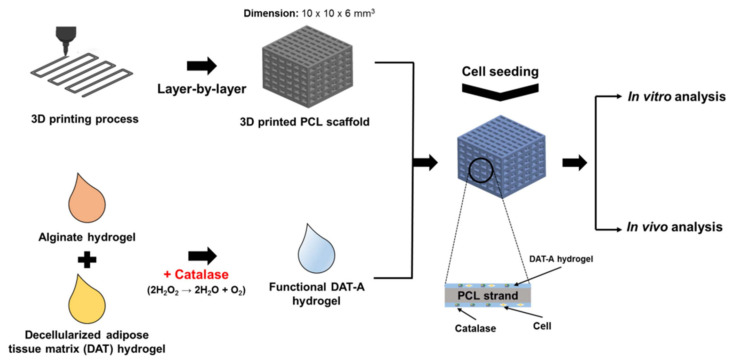
Schematic elucidation of the fabrication of functional 3D scaffold and 3D culture for in vitro and in vivo analysis [22].

**Figure 4 pharmaceutics-15-01700-f004:**
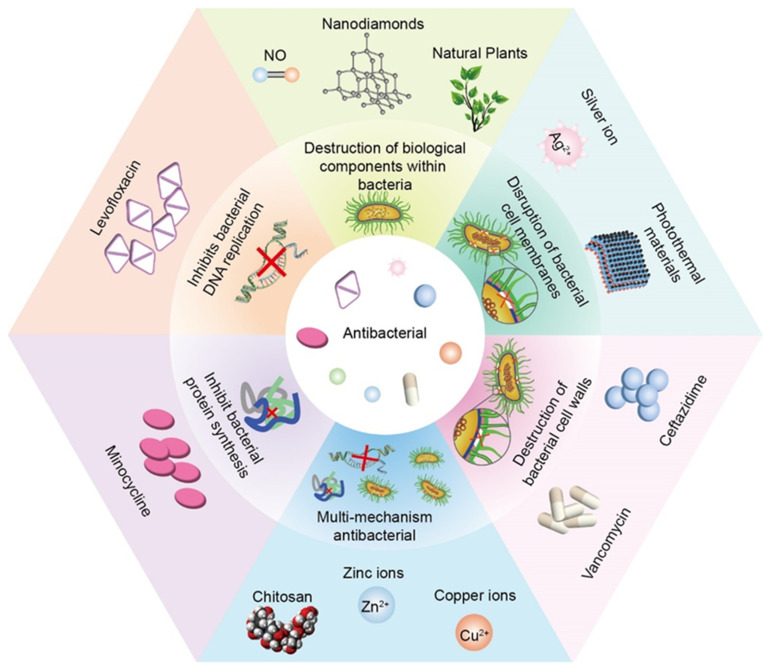
Different additives incorporated in the hydrogel according to the different antimicrobial mechanisms.

**Figure 5 pharmaceutics-15-01700-f005:**
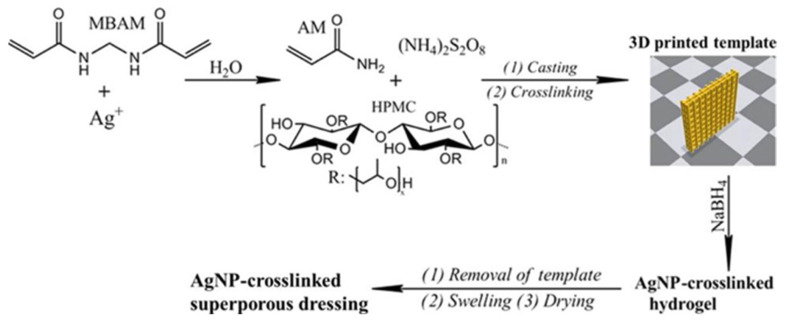
Preparation of the AgNP-Cross-Linked Superporous Hydrogel Dressing [85].

**Figure 6 pharmaceutics-15-01700-f006:**
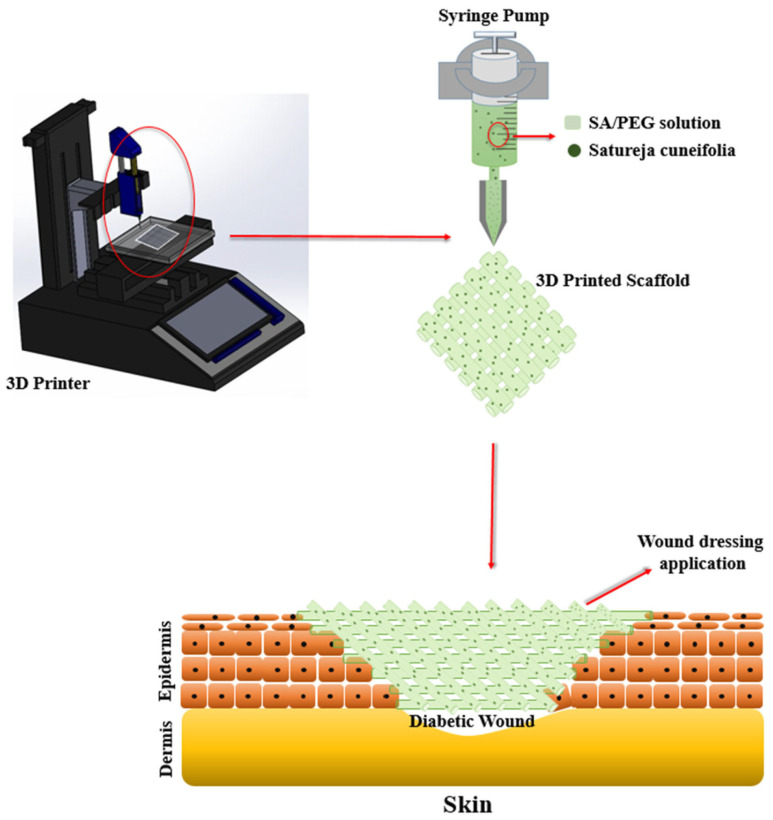
Schematic of 3D printer, fabrication of 3D-printed scaffold, and application to wound dressing [30].

**Figure 7 pharmaceutics-15-01700-f007:**
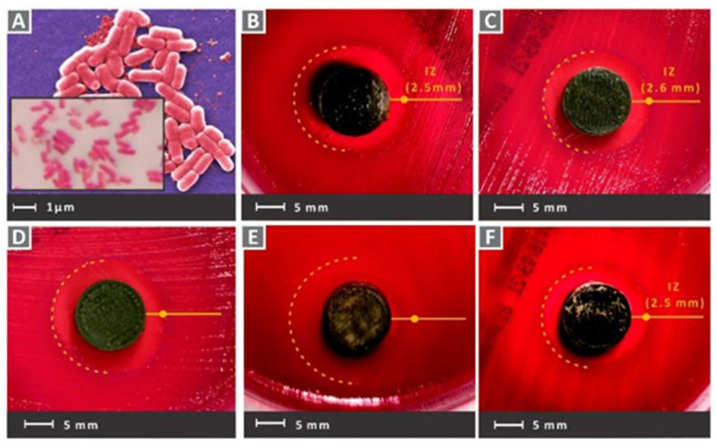
(**A**) Typical morphology of gram-negative *E. coli* captures of the tested samples for *E. coli* in the petri dishes after 24 h cultivation. (**B**) PA12/Cu_2_O 0.5 wt.%, (**C**) PA12/Cu_2_O 1.0 wt.%, (**D**) PA12/Cu_2_O 2.0 wt.%, (**E**) PA12/Cu_2_O 4.0 wt.%, (**F**) PA12/Cu_2_O 6.0 wt.% [83].

**Figure 8 pharmaceutics-15-01700-f008:**
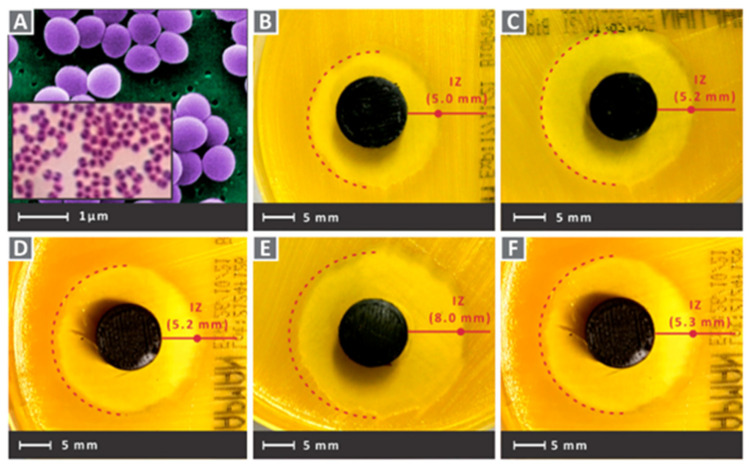
(**A**) Typical morphology of Gram-positive *S. aureus* captures of tested samples for Gram-positive *S. aureus* in Petri dishes after 24 h cultivation. (**B**) PA12/Cu_2_O 0.5 wt.%, (**C**) PA12/Cu_2_O 1.0 wt.%, (**D**) PA12/Cu_2_O 2.0 wt.%, (**E**) PA12/Cu_2_O 4.0 wt.%, and (**F**) PA12/Cu_2_O 6.0 wt.% [83].

**Figure 9 pharmaceutics-15-01700-f009:**
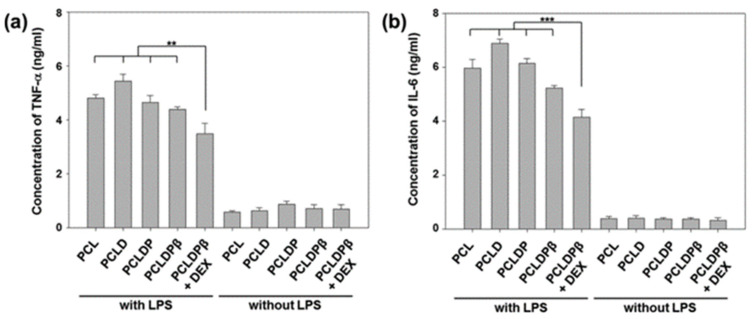
Anti-inflammatory response of various surface-modified membranes based on PCL fibrous membrane. The concentration of TNF-α (**a**) and IL-6 (**b**), secreted from the RAW 264.7 cells with and without LPS treatment for 24 h, was quantified by ELISA (** *p* < 0.01 and *** *p* < 0.001) [33].

**Figure 10 pharmaceutics-15-01700-f010:**
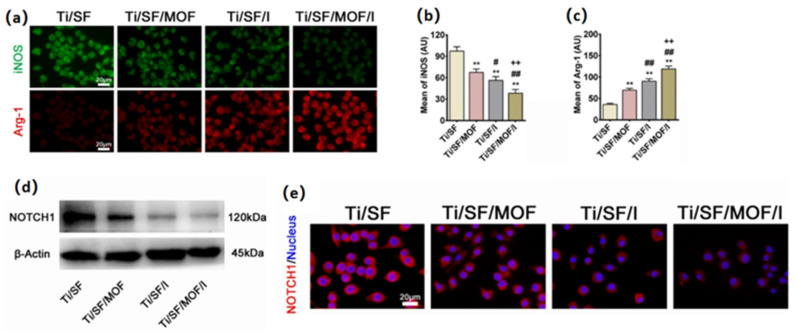
(**a**) Immunofluorescent staining of Raw264.7 cells after cultured for 4 days. (**b**,**c**) Quantitative analysis of iNOS and Arg-1. (**d**) The expression of Notch1 was detected by Western blotting. (**e**) Immunofluorescent staining of Notch1. (n = 3; # represent *p* < 0.05 when compared with Ti/SF, Ti/SF/MOF and Ti/SF/I, respectively; **, ## and ++ represent *p* < 0.01) [80].

**Figure 11 pharmaceutics-15-01700-f011:**
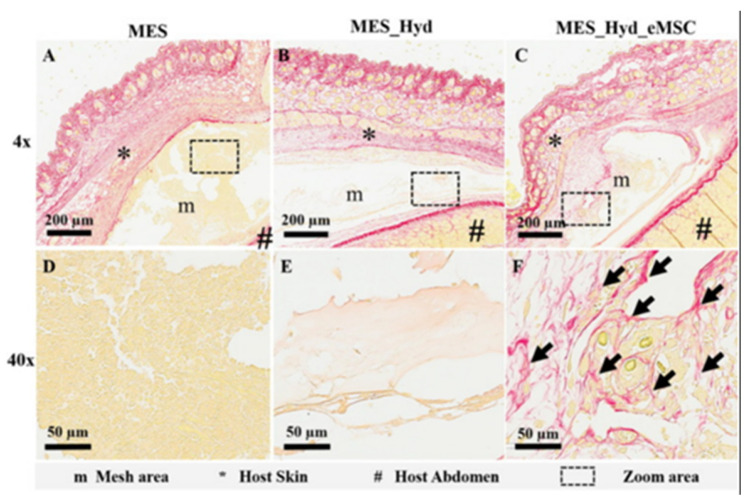
In vivo collagen deposition within the mesh after 1 week. Collagen deposition (black arrow) viewed by Picro Sirius red staining within mesh area in (**A**) MES, (**B**) MES-Hydrogel and (**C**) MES-Hy-eMSCs and (**D**–**F**) corresponding magnified view in dashed area [122].

**Figure 12 pharmaceutics-15-01700-f012:**
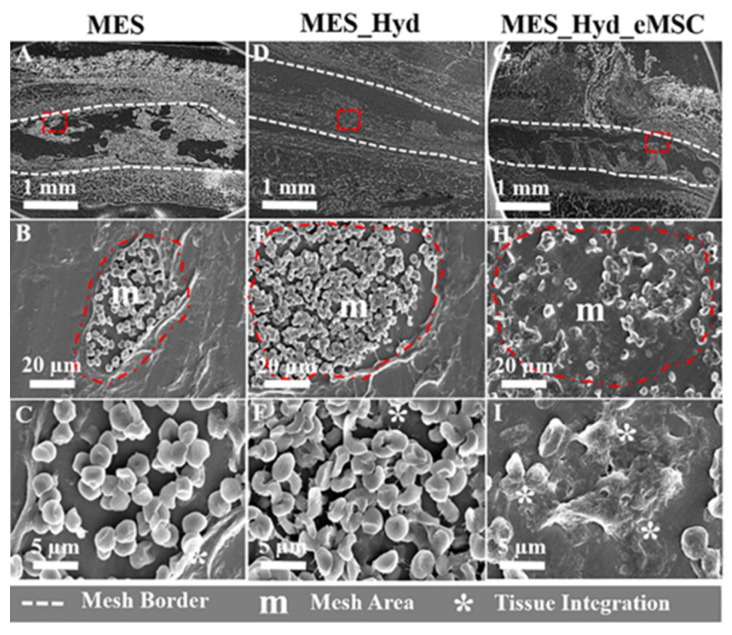
Fate of meshes after 1 week of in vivo implantation. SEM images show cross-sections of (**A**–**C**) MES (**D**–**F**) MES_Hydrogel and (**G**–**I**) MES_Hydrogel_eMSC constructs (within the red dashed area) 1 week after implantation in NSG mice; lower panels (**C**,**F**,**I**) show the morphology of reticular fibers (m), their interaction with the host tissue integration (white*) and the formation of new ECM [122].

**Table 1 pharmaceutics-15-01700-t001:** Different additives in hydrogels with different mechanism to methodology.

Function	Additive		Mechanism	Methodology	References
Vascularization	Growth factors	VEGF	VEGF can induce endothelial cell proliferation, promotes cell migration, and inhibits apoptosis. VEGF-induced angiogenesis with increased vascular permeability plays a critical regulatory role in the generation of blood vessels.	Wang et al. made BMP-2 and VEGF adsorbed onto silk fiber (SF) microspheres (diameter of 1.5 ± 0.3 μm) that were prepared using a co-flow capillary device. These microspheres were subsequently doped into the SF/nHAp scaffolds to provide regulated and controlled release.	[16]
		FGF-2	FGF-2 can stimulate VEGF expression, granulation tissue formation, and vascular maturation.	Xiong et al. used 3D printing to fabricate a gelatin-sulfonated silk composite scaffold (3DG-SF-SO_3_-FGF). The basic FGF-2 was incorporated in this scaffold by binding with a sulfonic acid group (SO_3_).	[17]
		PDGF	PDGF can recruit SMC/pericytes to immature vessels to stabilize and remodel them.	/	/
	Heparin and its derivatives	Heparin	Heparin can accelerate neovascularization by binding angiogenic factors, such as VEGF, and improve their stability.	An et al. coated poly-L-lysine (PLL) and heparin on the surface of 3D printed hydrogel scaffolds, which were made from cross-linking of GelMA and HAMA via electrostatic interactions between PLL and heparin.	[18]
		Heparan sulfate	Heparan sulfate can not only protect growth factors from degradation by proteases, but it also promote the binding of growth factors to their receptors, thus promoting growth factor activity.	Jiang et al. fabricated a scaffold with collagen and heparan sulfate	[19]
	Other materials	Angiogenic peptides	Angiogenic peptides are functionally equivalent to VEGF and can bind to the same receptors to initiate angiogenesis.	Wang et al. made a dual-delivery bone tissue engineering scaffold by low-temperature 3D printing of β-tricalcium phosphate and—osteogenic peptide (OP) containing water/PLGA/DCM emulsion—and coating AP on the scaffold surface.	[20]
		Desferrioxamine	Desferrioxamine can promote HIF1-α synthesis by simulating hypoxia, which was demonstrated to be essential for the regulation of genes related to angiogenesis. Desferrioxamine can lead to the activation of a cascade of pro-angiogenic genes, such as VEGF when it was activated.	Yan et al. designed a bionic degradable polycaprolactone (PCL) scaffold using 3D printing technology. It can control the release of DFO by surface degradation and layer-by-layer assembly techniques.	[21]
		Catalase	Catalase can break down hydrogen peroxide and produce oxygen, which not only reduces the damage caused by H_2_O_2_, but the oxygen that it produces also helps to induce angiogenesis	Rija et al. intercalated catalase in hydrogels to form functional decellularized adipose tissue-alginate (DAT-A) hydrogels using 3D printing technology.	[22]
		Decellularized extracellular matrix	The dECM plays a huge role in pro-vascularization with the removal of immunogenicity while retaining a large amount of nutrients such as large amounts of growth factors. It can facilitate the induced differentiation of pluripotent stem cells on dECM scaffolds into tissue-specific cell types	Kim et al. used S-dECM bioink to print and fabricate 3D pre-vascularized skin patches for wound healing with infusion of adipose-derived stem cells (ASCs) and endothelial progenitor cells (EPCs).	[23,24]
				Bejleri et al. fabricated a 3D bioprinted patch containing cECM for the delivery of pediatric cardiac progenitor cells hCPCs which is printed with bioinks consisting of cECM, hCPCs, and GelMA.	
Antibacterial	Photothermal materials		MXenes can kill microorganisms such as Gram-positive and negative bacteria by disrupting bacterial membranes directly through physical contact, especially when supplemented with NIR irradiation for higher antimicrobial efficiency	Nie et al. fabricated a personalized MXene composite hydrogel scaffold GelMA/β-TCP/sodium alginate (Sr2+)/MXene (Ti3C2) (GTAM) with both photothermal antibacterial and osteogenic capabilities using 3D printing technology.	[25]
	Antibiotic	Vancomycin + Ceftazidime	Vancomycin can exert a strong bactericidal ability by inhibiting cell wall biosynthesis through specific binding to the dipeptide d-Ala-d-Ala (AA) at the end of bacterial cell wall precursors. Ceftazidime can inhibit bacterial peptidoglycan synthesis by inhibiting penicillin-binding proteins, leading to cell wall instability and inhibition of synthesis and cell death.	Yu et al. fabricated a mesh-like PCL scaffold by using 3D printing technology and poly(lactic-co-glycolic-acid) (PLGA) nanofibers with a hybrid sheath core structure by using co-axial electrospinning technique. Subsequently, they fabricated this nanofiber membrane in two layers (electrospun PLGA/vancomycin/ceftazidime layer and a coaxially spun PLGA/BMP-2 layer).	[26]
		Levofloxacin	Levofloxacin can inhibit bacterial helicase activity, which leads to the inability of DNA to replicate and synthesize properly, ultimately causing bacterial death.	Sadaba et al. produced poly (lactic acid) (PLA) scaffolds with 3D printing technology and subsequently added polydopamine-coated BaSO_4_ particles within the scaffolds and adsorbed levofloxacin in it.	[27]
		Minocycline	Minocycline can inhibit protein synthesis in bacteria.	Martin et al. utilized 3D printing technology to prepare a PLA scaffold and combined the scaffold with collagen, minocycline and bio-inspired citrate hydroxyapatite nanoparticles (cHA).	[28]
	NO		NO can play a bactericidal role by damaging DNA, proteins and lipids of microorganisms.	Kabirian et al. designed a 3D printed small-diameter vascular graft (SDVG) which was printed from polylactic acid and coated with blending of 10 wt% S-nitroso-N-acetyl-D-penicillamine mixed in a polymer matrix consisting of poly (ethylene glycol) and polycaprolactone to achieve controlled release of NO.	[29]
	*Satureja cuneifolia*		*Satureja cuneifolia* (SC) is a natural aromatic plant which is rich in phenolic compounds with antimicrobial activity, which are strongly oxidizing and can cause protein coagulation and destroy bacterial proteins thus exerting antimicrobial effects.	Ilhan et al. used 3D printing technique to make SA/PEG composite scaffold and loaded methanol extract of SC into it.	[30]
	Nanodiamonds		The active oxygen-containing groups on the surface of NDS promote their interaction with cellular components to quickly kill Gram-positive and negative bacteria and prevent bacterial adhesion.	Rifai et al. fabricated selective laser melting titanium (SLM-Ti) scaffolds by using 3D printing and selective laser melting (SLM), and applied nanodiamond (ND) coating on the scaffolds for functionalization modification.	[31]
	Chitosan		Its antimicrobial activity results from the interaction of the positive charge it carries with ions on the surface of negatively charged cells. Such activity disrupts bacterial cell membranes, prevents the transport of cellular material, increase the internal osmotic pressure, and cause the rupture of microbial cells, among other things. Chitosan can interact with bacterial DNA and prevents DNA transcription, thus inhibiting microbial ribonucleic acid (RNA) synthesis.	Intini et al. used 3D printing technology to manufacture porous chitosan scaffolds with 200 μm inter-filament opening.	[32]
Immunomodulation	Adrenocorticos-teroids	Dexamethasone	It can reduce the expression of cyclooxygenase 2 (COX-2). It inhibits prostaglandin production, inflammatory signaling, and neutrophil and macrophage exudation and aggregation to the site of inflammation.	Lee et al. designed a robust and biodegradable 3D tubular scaffold by the combination of electrostatic spinning technique (ELSP) and 3D printing technique, and subsequently loaded DEX onto this scaffold using a mild surface modification reaction of PDA, polyethyleneimine (PEI), and carboxymethyl β-cyclodextrin (βCD).	[33]
		Prednisolone	It can inhibit the accumulation of inflammatory cells (including macrophages and leukocytes) at sites of inflammation, and their phagocytosis. It can prevent the release of lysosomal enzymes, and the synthesis and release of chemical mediators of inflammation.	Farto-Vaamonde et al. utilized two different ways to load prednisolone or dexamethasone into a 3D printed PLA scaffold. The first one is immersing the pre-printed 3D PLA scaffold in a prednisolone solution, which covers its surface with prednisolone and allows for rapid release to exert its antimicrobial properties. The second one is immersing the polylactic acid filament in dexamethasone solution to make the polylactic acid swells reversibly.	[34]
	Animal sources	Interleukin-4	IL-4 can activate the production of Th2 cytokine and thus promotes the polarization of M2-type macrophages. IL-4 has the ability to antagonize the Th1-driven pro-inflammatory immune response, as evidenced by the downregulation of the synthesis of pro-inflammatory cytokines such as IL-10 and TNF-α and the inhibition of pro-inflammatory chemokines.	Wang et al. used GelMA-Dextran (PGelDex) as bioink and incorporated both IL-4 loaded silver-coated gold nanorods (AgGNRs) and hMSCs.	[35]
		Ac2-26 peptide	It can inhibit tumor necrosis factor-α (TNF-α) production in monocytes, inhibit NF-κB signaling of the proinflammatory pathway, and promote phagocytosis of neutrophils.	Xu et al. fabricated a polylactic acid/4-arm polyethylene glycol hydrogel (PCL@tetra-PEG) composite scaffold with the encapsulation of Ac2-26 peptide.	[36]
	Plant sources	Curcumin	Its inhibitory effect on cytokine production and expression. Curcumin can exert anti-inflammatory activity in LPS/interferon-γ (IFN-γ) treated macrophages through several mechanisms, for example, by blocking nuclear factor-κB (NF-κB) and signal transducer and activator of transcription 1 (STAT1) signaling pathways, thereby inhibiting LPS-induced IL-6 expression in RAW264.7 cells. Flavin derivatives can also inhibit NO, TNF-α and IL-1β expression by suppressing the mitogen-activated protein kinase (MAPK)/NF-κB pathway in IFN-γ/LPS-stimulated macrophages.	Chen et al. made mesoporous CS (MesoCS/curcumin) scaffolds with curcumin. MesoCS nanoparticles were first prepared using a template followed by dissolving turmeric as a stock solution in 0.5 M NaOH to various concentration and finally mixing the scaffolds at the same 0.4 mL/g liquid/powder ratio to make the scaffolds.	[37]
		Isoflavones	It can eliminate free radicals produced by macrophages during inflammation, thereby reducing the release of various inflammatory mediators induced by ROS. It have ability to inhibit the phosphorylation of IκB, an inhibitor of NF-κB, thereby reducing the degradation of IκB, inhibiting NF-κB activation, and subsequently reducing the pro-inflammatory cytokine production among others.	Sarkar et al. loaded three soy isoflavones (Genistein, Daidzein, and Glycitein) in a 5:4:1 ratio mimicking their original ratios in soy on 3D printed TCP scaffolds with pores.	[38]
		Quercetin	It can induce phosphorylation of protein kinase B and promote signal transduction and nuclear translocation of transcriptional activator protein 6 (STAT6), it can increase the expression of the M2 phenotype upon binding to anti-inflammatory cytokine-related genes. Quercetin can block the NF-κB signaling pathway, thus promoting M2 polarization for anti-inflammatory repair effects.	Wang et al. fabricated layered micro/nano surfaces on the surface of Ti_6_Al_4_V implants using 3D printing technique, alkali heat treatment and hydrothermal treatment after deposition of titanium dioxide (TiO_2_) on this surface. Because of the excellent ability to chelate metal cations of quercetin, it can be adsorbed as a monomer on TiO_2_, effectively creating a quercetin coating on the surface of the 3D printed Ti_6_Al_4_V implant.	[39]
Regulation of collagen deposition	Collagen		Different types and ratios of collagen deposition play different roles in different periods of wound repair.	Martin et al. utilized 3D printing technology to prepare a PLA scaffold and combined the scaffold with collagen, minocycline and bio-inspired citrate hydroxyapatite nanoparticles (cHA).	[28]
	Extracellular Matrix		Extracellular matrix is mainly composed of various components such as collagen, non-collagen, and elastin, and collagen as the main component can not only give the extracellular matrix the function of directly regulating collagen deposition, but also avoid the degradation and shrinkage process caused by a single collagen.	Kim et al. used S-dECM bioink to print and fabricate 3D pre-vascularized skin patches for wound healing with infusion of ASCs and EPCs.	[23]
	Silk fiber		As one of the filamentous proteins, *Antheraea assama* SF can improve cell fate and promote ECM secretion owing to its Arg-Gly-Asp (RGD) content.	Bandyopadhyay et al. mixed a blend of SF from mulberry (*Bombyx mori*) and non-mulberry (*Antheraea assama*) silk with gelatin as inks for 3D bioprinting to simulate the extracellular microenvironment of the meniscus. Subsequently, EDC and NHS were chemically cross-linked to enhance their stability and mechanical strength.	[40]
	Curcumin		Curcumin has the function of inhibiting the NF-κβ signaling pathway, thus promoting osteoblast differentiation and the secretion of ECM.	Bose et al. fabricated a PCL-PEG+curcumin-coated TCP scaffold using 3D printing technology.	[41]
	Granulocyte colony-stimulating factor		As an anti-inflammatory protein, endometrial stem cell-derived G-CSF can reduce scar formation by reducing Gli2 protein expression levels to regulate collagen deposition and reduce endometrial fibrosis.	Wen et al. made G-CSF-loaded slow-release microsphere (G-CSF-SRM) hydrogel scaffolds with 3D printing technology.	[42]
	Polyvinylidene fluoride		Polyvinylidene fluoride (PVDF) has excellent piezoelectric properties, biocompatibility, thermal stability and resistance to chemical irritation. Electricity generated by piezoelectric materials can reverse the effects of the gradual weakening of bioelectric stimulation with wound healing which may cause disruption of gene regulation, leading to downregulation of the wound healing cascade and eventually disorganized deposition of collagen fibers and abnormal remodeling of the ECM.	Liang et al. fabricated a novel ZnO nanoparticle-modified PVDF/SA piezoelectric hydrogel 3D scaffold (ZPFSA). The incorporation of SA precisely reduces the hydrophobicity of PVDF and creates the possibility of preparing hydrogels from its 3D printing.	[43]

**Table 2 pharmaceutics-15-01700-t002:** Different functions of various ions in hydrogels.

Ions	Function	Mechanism	Methodology	Reference
Silicon ion	Vascularization	Promote the expression of angiogenesis-related factors in vascular endothelial cells. Stimulate in vitro migration, differentiation and tubule-like formation of vascular endothelial cells.	Use a novel calcium phosphate cement (CPC) as the basis for a scaffold that combined mesoporous silica (MS) with recombinant human bone morphogenetic protein-2 (rhBMP-2).	[77]
Magnesium ion	Vascularization	Stimulate the proliferation of HUVEC. Increase its response to some motogenic factors. Upregulate integrin function and thus promote endothelial cell migration.	Fabricated Mg-doped β-TCP (Mg-TCP) scaffolds by 3D printing and sintering, in which MgO was mixed in different ratios.	[78]
			Make Ta-PDA-Mg scaffolds by doping Mg^2+^ on the surface of 3D printed tantalum scaffolds by using the surface adhesion ability of polydopamine.	[79]
	Immunomodulation	Increase IκBα levels, and lead to reduced NF-κB activation and secretion of proinflammatory cytokines. Inhibit Notch1 protein expression to exert anti-inflammatory effects.	Load magnesium ions in the form of Mg-MOF-74 into 3D bioprinted scaffolds to exert anti-inflammatory effects in combination with icariin.	[80]
Zinc ions	Antibacterial	The negative charge of the bacterial film, an electrostatic reaction between the film and the oppositely charged ZnO nanoparticles leads to the rupture of the film eventually triggering bacterial death. ZnO nanoparticles can release free Zn ions to disrupt the internal ionic homeostasis of the bacterium and subsequently lead to the death of the bacterium.	Made yttrium oxide-stabilized zirconium oxide (3Y-ZrO_2_) nanopowder into 3Y-ZrO_2_ ceramic by 3D printing, mold plasticity, and sintering. Then the ISO resin was uniformly coated onto the ceramic. Subsequently, ZnO nanosolution was dropped on this ceramic to finally produce ZnO-ISO bilayer modified ceramics.	[81]
Copper ion	Vascularization	Stimulate endothelial cell proliferation and differentiation. Stimulates angiogenesis by mimicking hypoxia, stabilizing the expression of hypoxia-inducible factor (HIF-1a) and promoting the expression of VEGF.	Prepared a novel metal-organic framework, a β-tricalcium phosphate (Cu-TCPP-TCP) scaffold containing a copper coordinated tetrakis (4-carboxyphenyl) porphyrin (Cu-TCPP) nanosheet interfacial structure, by using 3D printing technology.	[82]
	Antibacterial	Highly toxic hydroxyl radicals produced by copper can oxidize with most bacterial macromolecules to exert antibacterial effects. Associate with affecting the bacterial outer membrane potential, leading to the rupture of the bacterial membrane, and eventually to bacterial death.	Used 3D printing technology to prepare nanocomposites of medical grade polyamide 12 (PA12) with copper oxide (cuprous oxide) in different ratios.	[83]
	Immunomodulation	Inhibit the activation of macrophages and suppress the synthesis and release of inflammatory factors.	Made Cu-BGC scaffolds by incorporating different concentrations of copper (0.781–25 mg/mL) into bioactive microcrystalline glass using 3D printing technology.	[84]
	Regulation of collagen deposition	Upregulate hypoxia-inducible factor-1 (HIF-1) expression and stimulates the expression of SOX-9, which leads to increased levels of COL II and ACAN expression.	Fabricated Cu-BGC scaffolds by doping different concentrations of copper (0.781–25 mg/mL) into bioactive microcrystalline glass with 3D printing technology.	
Silver ion	Antibacterial	Anchoring of AgNps on bacterial membranes leading to rupture of bacterial membranes and leakage of bacterial contents, resulting in bacterial cell death. Nanoparticles can penetrate into bacterial membranes and interact with biomolecules, resulting in bacterial dysfunction. AgNps may catalyze the reduction in O_2_ to reactive oxygen species (ROS), leading to downregulation of antioxidant enzyme expression, DNA damage and apoptosis.	Silver is mainly used in the form of metallic silver nanoparticles (AgNPs). Designed a superporous polyacrylamide (PAM)/Hydroxypropyl Methyl Cellulose (HPMC) hydrogel with antibacterial ability by using silver-ethylene interactions and 3D printing technology.	[85]

**Table 3 pharmaceutics-15-01700-t003:** Different cells in hydrogel.

Cells	Functions	Mechanism	Methodology	Merits	Reference
HUVECs	Vascularization	Prevent the need for pre-formed channels or growth factor-induced angiogenesis.	HUVECs was seeded in the Laponite (LAP) nanoclay with VEGF.	Low cost, ease of isolation, and strong angiogenic potential	[110]
MSCs	Vascularization	Increase the expression of vascular growth factors.	A 3D-printed GelMA hydrogel scaffold with a controlled internal structure which was filled with MSCs.	Immunomodulatory and tissue repair promoting effects Potential for multi-directional tissue differentiation The ability to secrete a variety of cytokines	[118]
	Immunomodulation	Promote paracrine activity during inflammation and induce a shift from an M1-type phenotype to an M2-type phenotype in macrophages.	A scaffold which was composed by the aloe vera–sodium alginate (AV-ALG) hydrogel-coated endometrial mesenchymal stem cells (eMSCs) and 3D melt electrospun wire nets.		[122]
	Regulation of collagen deposition	Inhibition of inflammation in scar tissue. Affect fibroblast activity in scar formation, and reduce scar formation by inhibiting the gene expression of TGF-β1 and decreasing type I collagen production. Play a positive role in collagen remodeling in scar tissue.			

## Data Availability

Not applicable.

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
