# Peer review of "Review on Additives in Hydrogels for 3D Bioprinting of Regenerative Medicine: From Mechanism to Methodology"

_pharmaceutics, 2023, doi:10.3390/pharmaceutics15061700_

Round 1

Reviewer 1 Report

“Review on additives in hydrogels for 3D bioprinting of regenerative medicine: from mechanism to methodology” This review is comprehensive and good paper, but needs some major revisions to improve the quality, as follows:

1. authors need to combine the roles of many biological factors, and ions, to explain the effects.

2. authors should discuss the merits and demerits of each cell type, table may be helpful.

3. Some of the recent papers on 3D bioprinting works should be referenced and discussed within text, as follows:

- Tingting Weng et al. 3D bioprinting for skin tissue engineering: Current status and perspectives. 2021.

- Yilmaz et al. Bioprinting: A review of processes, materials and applications, 2021.

- Muthu Parkkavi Sekar et al. Current standards and ethical landscape of engineered tissues—3D bioprinting perspective. 2021.

Agarwal et al. Current Developments in 3D Bioprinting for Tissue and Organ Regeneration–A Review, 2020.

- Patricia Santos-Beato et al.Biofabrication of the osteochondral unit and its applications: Current and future directions for 3D bioprinting. 2022.

4. perspective part needs improvement; explaining future directions, current hurdles, etc.

n/a

Author Response

Dear reviewer,

Thank you for the time and energy you put into reviewing the manuscript.  I wish you all the best. Here are my responses to the referees’comments.

  1. I have followed the suggestions and described the various ion-cell interactions to clarify the role played by ions.
  2. I have made a table that discusses the functions of the various cells, the mechanisms by which they work and their advantages.
  3. After carefully reading the literature recommended by the reviewers, I revised the article and cited all the recommended literature.
  4. I have revised the viewpoint section to enrich the difficulties encountered so far and the future research directions.

The revised manuscript please see the attachment.

Thank you again for your all work for reviewing!

Reviewer 2 Report

This paper presents the additives in hydrogels for 3D bioprinting of regenerative medicine. I find that the manuscript is well written and would attract a great deal of attention when published. However, before considering publication, it should be further improved to include more balanced and in-depth information.

1.     There are too many typos. For example, the authors did not use superscript or subscript such as Mg (NO3)2, Mg2+.

2.     Is this paper short or mini review or invited paper from editor? Generally, there are lack of contents of this paper. For example, the authors should explain the reason for dividing 4 kinds of additives in introduction.

3.     Same contents are duplicated, even though the purpose were different such as Si ion, curcumin and so on.

4.     The title of this paper includes ‘from mechanism to methodology’, thus additional figures for explanation of mechanism of each 4 purposes.

5.     I think additional tables can be added for providing more information for readers. For example, there are simple examples of additives for vascularization. But the author can additional information for finding additives easily.

6.     And It is strongly recommended to add additional reference as below. https://doi.org/10.1002/advs.202300816

https://doi.org/10.1002/adfm.202206863

doi: 10.18063/IJB.v4i1.126

https://doi.org/10.1177/20417314211057236

7.     Even though this paper is review paper, but too many sentences were duplicated with other papers. I am attaching the document for checking redundancy. Please check it.

..

Author Response

Dear reviewer,

Thank you for the time and energy you put into reviewing the manuscript.  I wish you all the best. Here are my responses to the referees’ comments.

  1. I've revised the typos and amended the superscript or subscript error.
  2. I have explained the reason why the various additives are divided into 4 main categories in the introduction.
  3. I have made a table of repetitions such as silicon ions, copper ions, etc. so that the reader can distinguish the different functions of the same additive.

     4+5. I have created a detailed table explaining the role of the different additives and their mechanisms, as well as the methods used by researchers, so that readers can more easily find the right additive. A brief description of the mechanism of each 4 purposes is also given.

  1. After carefully reading the literature recommended by the reviewers, I revised the article and cited all the recommended literature.
  2. I am so sorry that I don't see the document for checking redundancy which is attached.Would you mind upload it again?

The revised manuscript please see the attachment.

Thank you again for your all work for reviewing!

Reviewer 3 Report

The manuscript entitled, ‘Review on additives in hydrogels for 3D bioprinting of regenerative medicine: from mechanism to methodology’ discussed 3D printed hydrogels for biomedical applications. The article should be modified according to the following points;

1.      Author should emphasize the way how this review is different from others.

2.      According to materials point of view it will be better if some viscosity of printing ink or their flow behavior is discussed with proper references.

3.      Heading “2.4 Cell” should be modified with more promising heading.

4.      A table of different ions and their recurring activity would be more catchy for this review.

5.      Some articles have major significance for your references:

(a)    Ganguly, S., & Margel, S. (2022). 3D printed magnetic polymer composite hydrogels for hyperthermia and magnetic field driven structural manipulation. Progress in Polymer Science, 101574.

(b)   Ganguly, S., Das, T. K., Mondal, S., & Das, N. C. (2016). Synthesis of polydopamine-coated halloysite nanotube-based hydrogel for controlled release of a calcium channel blocker. RSC advances6(107), 105350-105362.

(c)    Bhowmik, M., Kumari, P., Sarkar, G., Bain, M. K., Bhowmick, B., Mollick, M. M. R., ... & Chattopadhyay, D. (2013). Effect of xanthan gum and guar gum on in situ gelling ophthalmic drug delivery system based on poloxamer-407. International journal of biological macromolecules62, 117-123.

(d)   Sarkar, G., Orasugh, J. T., Saha, N. R., Roy, I., Bhattacharyya, A., Chattopadhyay, A. K., ... & Chattopadhyay, D. (2017). Cellulose nanofibrils/chitosan based transdermal drug delivery vehicle for controlled release of ketorolac tromethamine. New Journal of Chemistry41(24), 15312-15319.  

Author Response

Dear reviewer,

Thank you for the time and energy you put into reviewing the manuscript.  I wish you all the best. Here are my responses to the referees’ comments.

  1. I have highlighted how this review differs from other reviews and the advantages of this review.
  2. I sincerely hope you'll excuse m Since many of the additives do not change the hydrodynamics and viscosity of the material, I did not focus on the mechanical properties of the material so much. Instead, I have emphasized the slow release and local release of the materials according to their properties.
  3. I have revised the title from “cell”to “Vascular-Promoting Cell”.
  4. I have designed a table to describe the functions and mechanisms of the different ions to make it easier for the reader to choose the right ion.
  5. After carefully reading the literature recommended by the reviewers, I revised the article and cited all the recommended literature.

The revised manuscript please see the attachment.

Thank you again for your all work for reviewing!

Reviewer 4 Report

The manuscript submitted by Fang et al is a very up-to-date review that will certainly be useful to many readers working on the development of scaffolds for regenerative medicine.

The review is clearly structured, well written and the information is thoroughly analyzed. It merits to be published in Pharmaceutics. However, I would recommend a minor revision before accepting the article:

1.       There is no figure caption to Figure 4. Please provide.

2.       An explanation is required for each figure that has been reproduced or adapted from the work cited. In addition, for the reproduced figures, information should be provided on the conditions of their reproduction (with the permission of the publisher or under an open license).

Author Response

Dear reviewer,

Thank you for the time and energy you put into reviewing the manuscript.  I wish you all the best. Here are my responses to the referees’ comments.

  1. I have completed the figure caption to Figure 4.
  2. I have resolved copyright issues about figures. 

The revised manuscript please see the attachment.

Thank you again for your all work for reviewing!

Round 2

Reviewer 2 Report

.

Reviewer 3 Report

It can be published in its present state.